# FAK activity sustains intrinsic and acquired ovarian cancer resistance to platinum chemotherapy

Carlos J Diaz Osterman[1†], Duygu Ozmadenci[1†], Elizabeth G Kleinschmidt[1†], Kristin N Taylor[1†], Allison M Barrie[1†], Shulin Jiang[1†], Lisa M Bean[1†], Florian J Sulzmaier[1†], Christine Jean[1‡], Isabelle Tancioni[1], Kristen Anderson[1], Sean Uryu[1], Edward A Cordasco[1], Jian Li[2], Xiao Lei Chen[2], Guo Fu[2], Marjaana Ojalill[3], Pekka Rappu[3], Jyrki Heino[3], Adam M Mark[4], Guorong Xu[4], Kathleen M Fisch[4], Vihren N Kolev[5], David T Weaver[5], Jonathan A Pachter[5], Balázs Győrffy[6,7], Michael T McHale[1], Denise C Connolly[8], Alfredo Molinolo[9], Dwayne G Stupack[1*], David D Schlaepfer[1*]

[1]Department of Obstetrics, Gynecology and Reproductive Sciences, Moores UCSD Cancer Center, La Jolla, United States; [2]State Key Laboratory of Cellular Stress Biology, Innovation Center for Cellular Signaling Network, School of Life Sciences, Xiamen University, Xiamen, China; [3]Department of Biochemistry, University of Turku, Turku, Finland; [4]Department of Medicine, UCSD Center for Computational Biology & Bioinformatics, La Jolla, United States; [5]Verastem Oncology, Needham, United States; [6]Institute of Enzymology, Hungarian Academy of Sciences, Budapest, Hungary; [7]2nd Department of Pediatrics, Semmelweis University, Budapest, Hungary; [8]Fox Chase Cancer Center, Philadelphia, United States; [9]Department of Pathology, Moores UCSD Cancer Center, La Jolla, United States

*For correspondence:
dstupack@ucsd.edu (DGS);
dschlaepfer@ucsd.edu (DDS)

†These authors contributed equally to this work

Present address: ‡INSERM UMR1037, Centre de Recherches en Cancérologie de Toulouse, Toulouse, France

**Abstract** Gene copy number alterations, tumor cell stemness, and the development of platinum chemotherapy resistance contribute to high-grade serous ovarian cancer (HGSOC) recurrence. Stem phenotypes involving Wnt-β-catenin, aldehyde dehydrogenase activities, intrinsic platinum resistance, and tumorsphere formation are here associated with spontaneous gains in _Kras_, _Myc_ and _FAK_ (KMF) genes in a new aggressive murine model of ovarian cancer. Adhesion-independent FAK signaling sustained KMF and human tumorsphere proliferation as well as resistance to cisplatin cytotoxicity. Platinum-resistant tumorspheres can acquire a dependence on FAK for growth. Accordingly, increased FAK tyrosine phosphorylation was observed within HGSOC patient tumors surviving neo-adjuvant chemotherapy. Combining a FAK inhibitor with platinum overcame chemoresistance and triggered cell apoptosis. FAK transcriptomic analyses across knockout and reconstituted cells identified 135 targets, elevated in HGSOC, that were regulated by FAK activity and β-catenin including Myc, pluripotency and DNA repair genes. These studies reveal an oncogenic FAK signaling role supporting chemoresistance.
DOI: https://doi.org/10.7554/eLife.47327.001

## Introduction

Ovarian carcinoma is the most lethal gynecologic malignancy in the United States (*Siegel et al., 2018*). High-grade serous ovarian carcinoma (HGSOC), the most prevalent histologic tumor subtype (*Matulonis et al., 2016*), is treated with a combination of cytoreductive surgery and carboplatin (DNA damage generation) and paclitaxel (microtubule-stabilizing drug) chemotherapy. Cure is highly

**eLife digest** Ovarian cancer is one of the deadliest types of cancer in women. There are two main reasons for the aggressiveness of this cancer. First, ovarian cancer cells can spread to other parts of a woman's body before she has been diagnosed, where the cells grow as tiny clumps or spheres of tumor cells, also called tumorspheres. Second, in the majority of patients, some ovarian cancer cells will develop resistance to the chemotherapy used. It is not clear exactly how these tumor cells become resistant to therapy. One way in which cells could do this is by gaining extra copies of genes that remove toxic substances or repair DNA, which help them withstand the therapy.

Here, Osterman, Ozmadenci, Keinschmidt, Taylor, Barrie, Jiang, Bean, Sulzmaier et al. set up a new experimental method to study how some ovarian cancer cells resist chemotherapy. Comparing ovarian cancer cells from mice at early and late stages of the disease showed that the later-stage, more aggressive cells had more genetic changes. One of these changes affected the gene for a protein called FAK, which was found to have more copies than normal. The FAK protein is an enzyme that helps cancer cells move around. In cells from mice with late-stage cancer, FAK was over-active and present at high levels. When these cells grew as tumorspheres, the tumors were more resistant to chemotherapy than their early-stage counterparts. In patients who have received chemotherapy, surviving tumor cells also exhibit high levels of FAK activity.

Human ovarian cancer cells that are resistant to chemotherapy can be grown into tumors in mice, where they retain their resistance to chemotherapy. However, if chemotherapy is combined with a drug that targets the FAK enzyme, the tumors shrink. This experiment highlights a possible weak spot of these tumor cells. To understand how FAK makes ovarian cancer cells resistant to chemotherapy, Osterman et al. deleted the gene for FAK from the cells and then looked at how this changed the levels of activation of different genes. They found that, in addition to its effects on cell movement, FAK also activated a group of genes that increase resistance to chemotherapy and repair damaged DNA.

This better understanding of how ovarian cancer cells resist chemotherapy could lead to new therapies. In particular, there is now a clinical trial for women with chemo-resistant ovarian cancer in which standard chemotherapy is combined with an inhibitor of the FAK protein.
DOI: https://doi.org/10.7554/eLife.47327.002

dependent on elimination of microscopic disease (*Narod, 2016*). Approximately 80% of patients with HGSOC exhibit serial disease recurrence, develop resistance to platinum chemotherapy, and die (*Bowtell et al., 2015*). Although platinum chemotherapy is effective at creating DNA damage and triggering cell apoptosis, subpopulations of tumor cells can survive this stress (*Pogge von Strandmann et al., 2017*).

Tumor sequencing has revealed complexity and heterogeneity among HGSOC (*Cancer Genome Atlas Research Network, 2011*). DNA breakage and regions of chromosomal gain or loss are common (*Patch et al., 2015*). Gains at 8q24 occur in most HGSOC tumors and encompass the *MYC* oncogene at 8q24.21 (*Gorringe et al., 2010*). Although *MYC* expression is frequently high in HGSOC, the clinical significance remains unclear. *MYC* supports pluripotent stem cell generation and contributes to chemoresistance (*Fagnocchi and Zippo, 2017*; *Kumari et al., 2017*; *Li et al., 2019*).

Myc protein expression is regulated by Wnt/β-catenin signaling, which is both essential for embryonic development and activated in many tumors (*Shang et al., 2017*). Wnt and Myc fall within the 10 most prevalent signaling pathways in cancer (*Sanchez-Vega et al., 2018*). Wnt signaling is tightly regulated by the stability, subcellular localization, and transcriptional activity of β-catenin, which supports cancer stem cell (CSC) survival and chemoresistance (*Condello et al., 2015*; *Nagaraj et al., 2015*). Platinum can, paradoxically, also select for ovarian cancer 'stemness' through undefined mechanisms (*Wiechert et al., 2016*). Increased aldehyde dehydrogenase (ALDH) activity, arising from elevated expression of a family of cellular detoxifying enzymes, is one hallmark of ovarian CSCs (*Raha et al., 2014*; *Silva et al., 2011*). Culturing cells as tumorspheres in vitro increases chemotherapy resistance, ALDH expression, cell de-differentiation and stemness (*Shah and Landen, 2014*;

*Malta et al., 2018*). Notably, HGSOC dissemination involves tumorsphere growth and survival within ascites (*Pogge von Strandmann et al., 2017*).

The *PTK2* gene at 8q24.3, encoding focal adhesion kinase (FAK), is frequently amplified in breast, uterine, cervical, and ovarian tumors (*Kaveh et al., 2016*). FAK is a cytosolic tyrosine kinase canonically activated by matrix and integrin receptors controlling cell motility (*Mitra et al., 2005*). Autophosphorylation at tyrosine 397 (pY397) is a hallmark of FAK activity (*Kleinschmidt and Schlaepfer, 2017*). HGSOC tumors with *PTK2* gains exhibit elevated FAK expression and FAK Y397 phosphorylation (*Cancer Genome Atlas Research Network, 2011*; *Zhang et al., 2016*). Metastatic HGSOC tumor micro-environments are enriched with matrix proteins that are FAK activators (*Pearce et al., 2018*). FAK knockdown and FAK inhibitor studies support an important role for FAK in promoting invasive tumor growth (*Ward et al., 2013*; *Tancioni et al., 2014*), yet the targets downstream of FAK are varied and may be tumor or stroma context-dependent (*Sulzmaier et al., 2014*; *Haemmerle et al., 2016*). Interestingly, phenotypes associated with FAK knockout can be distinct from FAK inhibition, since kinase-inactive FAK retains important scaffolding roles (*Lim et al., 2008*).

Several ATP-competitive FAK inhibitors have been developed. Acceptable Phase I safety profiles in patients with advanced solid tumors (*Jones et al., 2015*; *Soria et al., 2016*; *Hirt et al., 2018*) have enabled current Phase II combinatorial clinical trials with FAK inhibitors in pancreatic, mesothelioma, and non-small cell lung carcinoma (NCT02758587 and NCT02546531). In ovarian and prostate carcinoma preclinical models, FAK inhibition (VS-6063, defactinib) enhanced taxane-mediated tumor apoptosis (*Kang et al., 2013*; *Lin et al., 2018*). While inhibitors of FAK and Myc exhibit combinatorial activity in promoting HGSOC cell apoptosis in vitro (*Xu et al., 2017*), it remains uncertain whether gains in 8q24 encompassing *PTK2* are associated with specific HGSOC cell phenotypes or responses to therapy, as determinants of FAK pathway dependence in tumors remain unknown.

Herein, we molecularly characterize a new murine model of ovarian cancer that displays spontaneous gains in the <u>K</u>ras, <u>M</u>yc, and <u>F</u>AK genes among other striking similarities to HGSOC phenotypes. By using a combination of genetic FAK knockout and rescue, pharmacological inhibition, sequencing and bioinformatics, we identify a non-canonical FAK activity-dependent linkage to β-catenin leading to differential mRNA target expression of Myc and other targets supporting pluripotency and DNA repair. Our studies linking intrinsic FAK activity to platinum resistance support the combinatorial testing of FAK inhibitors for recurrent ovarian cancer.

## Results

### A new in vivo evolved murine epithelial ovarian cancer model

HGSOC is characterized by p53 inactivation and genomic copy number alterations (CNAs), though no preclinical models exist to study cell phenotypes associated with ovarian tumor CNAs. Murine ID8 cells, are spontaneously-immortalized clonal ovarian epithelial cells that form slow-growing tumors in C57Bl/6 mice (*Roby et al., 2000*). ID8 cells do not contain common oncogenic mutations and express wild type p53. Targeted p53 inactivation promotes ID8 tumor growth and sensitivity to platinum chemotherapy (*Walton et al., 2016*; *Walton et al., 2017*). Passage of ID8 cells through C57Bl/6 mice can enhance ID8 tumorigenic potential via undefined mechanisms (*Clark et al., 2016*; *Mo et al., 2015*; *Ward et al., 2013*).

We previously isolated aggressive ID8-IP cells, lethal in mice within 40 days (*Ward et al., 2013*), via early recovery of ascites-associated cells and anchorage-independent expansion ex vivo (*Figure 1A*). Total exome sequencing (90% of exons sequenced at 100X) of ID8 and ID8-IP cells revealed 19619 shared, 29373 ID8 unique, and 11800 ID8-IP unique gene variants. However, less than 1% of exon variants identified were detected by RNA sequencing (~60 million clean reads/replicate). No equivalent mutations were found in COSMIC, the Catalogue of Somatic Mutations in Cancer. In addition to non-synonymous mutations previously identified in ID8 cells (*Walton et al., 2016*), we detected two additional changes in *Hjurp*. In ID8-IP cells, new mutations were identified in *Xxylt1* and *Atxn10*. Overall, the mutational burden within both ID8 and ID8-IP cells is low.

To determine if genetic copy number alterations underlie ID8-IP phenotypes, exome sequencing read values and bioinformatic analyses were used to map sites of DNA gains or loss across chromosomes using ID8 as a reference (*Figure 1—source data 1*). Gains in murine chromosome cytoband regions 6qD1-G3, 15qD3-F3, and 15qA1-D3 were present in ID8-IP cells (*Figure 1B*, *green circles*).

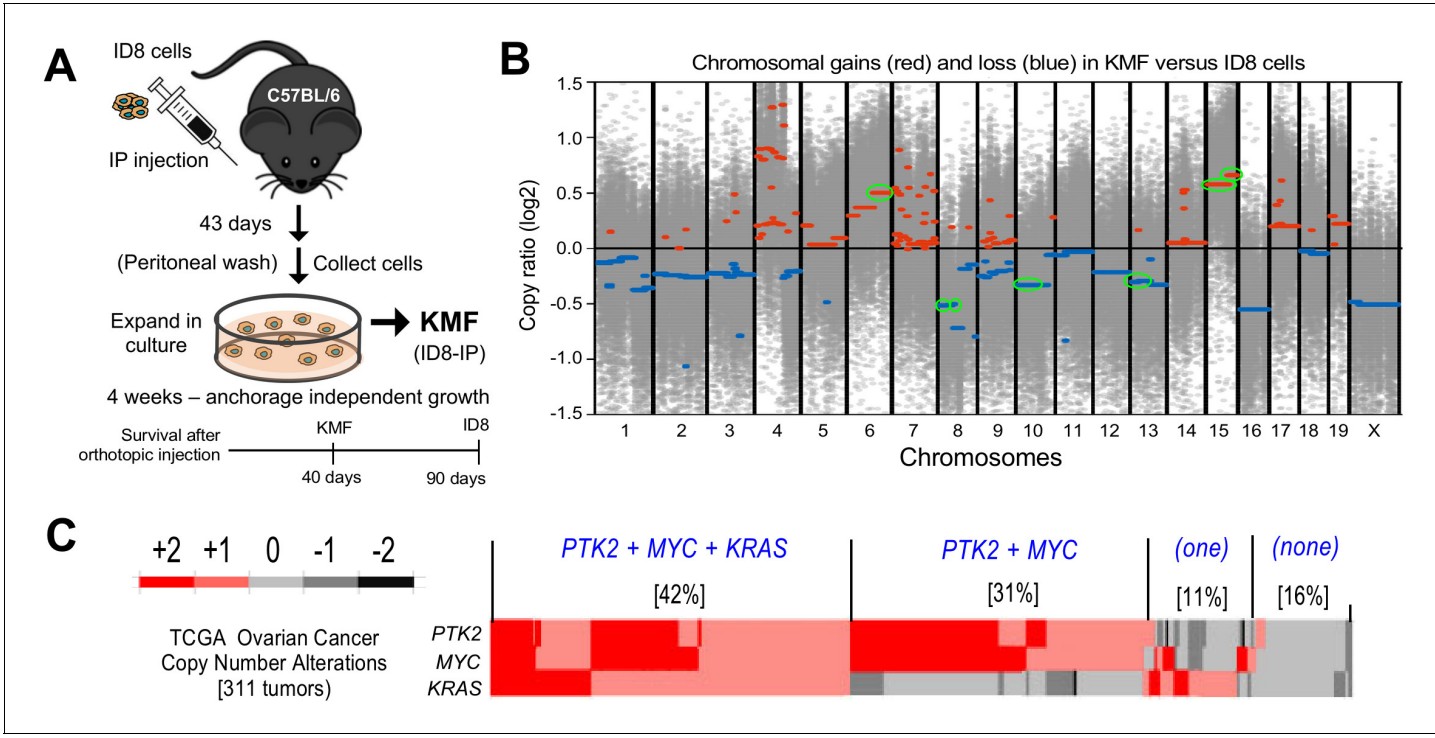

**Figure 1.** Spontaneous copy number gains in genes for _Kras_, _Myc_, and _FAK_ (_Ptk2_) in a new murine model (KMF) of ovarian cancer. (**A**) Schematic summary of KMF cell isolation by in vivo selection for aggressive ID8 growth in C57Bl/6 mice and expansion of cells as tumorspheres. (**B**) Whole-genome copy number ratio (log2) determined from ID8 and KMF exome sequencing. Gains (red) and losses (blue) are denoted across chromosomes. Circled regions (green) highlight shared genomic copy alterations between KMF and HGSOC (_Table 1_). (**C**) Heat map showing genomic copy number alterations encompassing _KRAS_, _MYC_, and _PTK2_ genes in HGSOC patients (TCGA, 311 tumors). Percentage of tumors with +1 or +2 copy number gains per group are indicated.

DOI: https://doi.org/10.7554/eLife.47327.003

The following source data and figure supplements are available for figure 1:

**Source data 1.** ID8 and KMF copy number alterations determined from exome sequencing.

DOI: https://doi.org/10.7554/eLife.47327.006

**Figure supplement 1.** Analysis of _PTK2_ mRNA and FAK protein expression as a function of genomic copy number.

DOI: https://doi.org/10.7554/eLife.47327.004

**Figure supplement 2.** Elevated _PTK2_ mRNA is associated with a poor prognosis in ovarian cancer.

DOI: https://doi.org/10.7554/eLife.47327.005

These correspond to human cytobands 12p12.1, 8q24.2, and 8q24.3. The latter two represent one of the most amplified regions in HGSOC (_Cancer Genome Atlas Research Network, 2011_; _Li et al., 2019_). The gain in cytoband 15qA1-D3 is in addition to chromosome 15 polyploidy detected by ID8 karyotyping (_Roby et al., 2000_). Notably, common gene gains in ID8-IP and HGSOC include _Kras_, _Myc_, and _Ptk2_ (encoding FAK) that support proliferation, stem cells, and adhesion signaling, respectively (_Table 1_). Herein, these ID8-IP cells will be termed KMF to denote gains in _Kras_, _Myc_, and _FAK_ genes. Murine KMF cells contain several gains or losses in genes common to the top 20 set of genes altered in HGSOC (_Table 1_).

## MYC and PTK2 associations in HGSOC

In HGSOC patients, simultaneous gains in _KRAS_, _MYC_, and _PTK2_ gains co-occur in 42% of tumors; _PTK2_ and _MYC_ co-occur in an additional 32% of HGSOC patients (_Figure 1C_). Thus, more than 70% of HGSOC tumors contain combined gains at _PTK2_ and _MYC_ loci. _PTK2_ copy number gains are linearly proportional to _PTK2_ mRNA ($R^2$ = 0.66) and FAK protein ($R^2$ = 0.61) levels in HGSOC tumors (_Figure 1C—figure supplement 1_). Elevated _PTK2_ mRNA levels in HGSOC are associated with decreased patient relapse-free survival (n = 1435, p=0.0009, and hazard ratio = 1.25) (_Figure 1—figure supplement 2A_). Bioinformatic analyses identified a set of 36 genes on different chromosomes

**Table 1.** Shared copy number alterations between KMF and the top 20 most significant gene gains and losses in HGSOC.

| Murine Cytoband | Human Cytoband | Gain/ Loss | Genes in common murine-human loci | Pathway/Role |
| --- | --- | --- | --- | --- |
| 6qD1-G3 | 12p12.1 | Gain | KRAS | Proliferation |
| 15qA1-D3 | 8q24.21, 8q24.3 | Gain | MYC, PTK2 | Stem Cell, Adhesion |
| 15qD3-F3 | 8q24.3 | Gain | RECQL4 | DNA Repair |
| 8qA1.1–1.3 | 8p23.3 | Loss | TUSC3 | Tumor Suppressor |
| 8qB1.1–1.2 | 4q34.3 | Loss | IRF2 | Interferon Response |
| 10qA1-D1 | 19p13.3 | Loss | TJP3 | Tight Junction |
| 13qB3-D2.3 | 5q11.2, 5q13.1 | Loss | MAP3K1, FOXD1, PIK3R1 | MAPK, Cell Cycle P85-PI3-kinase |

DOI: https://doi.org/10.7554/eLife.47327.007

in HGSOC that exhibit a significant and at least a two-fold change in tumors with elevated *PTK2*. This 36 gene set was associated with a significant shorter time to relapse (n = 575, p=0.0024, hazard ratio = 1.37) (*Figure 1—figure supplement 2B,C*). Together, these results support the importance of *PTK2* gains as a marker for poor prognosis.

## KMF cells exhibit enhanced CSC phenotypes and cisplatin resistance

ID8 cells exhibit an epithelial-like morphology and poor growth as colonies in semi-solid methylcellulose media. KMF cells exhibit a mesenchymal morphology, form foci in two-dimensional (2D) cell culture, and readily form 3D colonies in methylcellulose (*Figure 2A,B*). When grown in serum-free supplement-enhanced tumorsphere media (PromoCell) under anchorage-independent conditions for 5 days, ID8 and KMF cells remain ~95% viable and can be analyzed for signaling differences. By immunoblotting, KMF tumorspheres expressed elevated FAK, increased FAK Y397 phosphorylation, and decreased E-cadherin and β-catenin protein levels relative to ID8 cells and normalized to actin (*Figure 2C*). Treatment of KMF tumorspheres with a glycogen synthase kinase-3 inhibitor (GSK3i) increased β-catenin protein and nuclear transcriptional activity whereas GSK3i addition had no effects on β-catenin levels or activity in ID8 cells (*Figure 2D*). Additionally, greater than 10% of KMF tumorspheres possessed high ALDH activity with less than 1% of ID8 cells being ALDH-positive (*Figure 2E*). These results support the notion that KMF cells have gained enhanced CSC characteristics.

To determine if ID8 and KMF cells possess distinct transcriptional signatures, RNA sequencing was performed (*Figure 2—source data 1*). Using FPKM (Fragments Per Kilobase of transcript per Million mapped reads) threshold values greater than one, 10800 shared, 744 ID8 enriched, and 402 KMF elevated transcripts were identified (*Figure 2F*). Top 20 Reactome signaling pathways upregulated in KMF cells include *Cell Cycle Control*, *Mitotic Checkpoint*, *DNA Repair, and Rho GTPase* signaling (*Figure 2G*). Elevated cell cycle mRNA levels are consistent with enhanced KMF tumorsphere growth. However, elevated levels of mitotic checkpoint inhibitors such as p21CIP1 were also constitutively and highly expressed in KMF cells (*Figure 2H*). Deregulated p21CIP1 levels can occur in p53-deficient cells (*Georgakilas et al., 2017*), but no mutations in p53 were detected by KMF exome sequencing and steady-state levels of KMF p53 protein are low. As DNA repair pathway targets are also increased in KMF cells (*Figure 2G*), ID8 and KMF tumorsphere viability was measured after exposure to different concentrations of cisplatin (CP) over 5 days (*Figure 2I*). KMF cells possessed increased intrinsic resistance to CP cytotoxicity with greater than a 10-fold difference in EC50 values compared to ID8. Taken together, this new KMF model exhibits noteworthy phenotypic similarities to drug-resistant HGSOC.

## Sustained FAK Y397 phosphorylation (pY397) in patient ovarian tumors surviving neoadjuvant chemotherapy

A small subset of HGSOC patients are treated with neoadjuvant carboplatin and paclitaxel chemotherapy to reduce tumor burden prior to undergoing surgery (*Matulonis et al., 2016*). However, some tumor cells, such as CSCs, escape CP-mediated apoptosis and survive chemotherapy

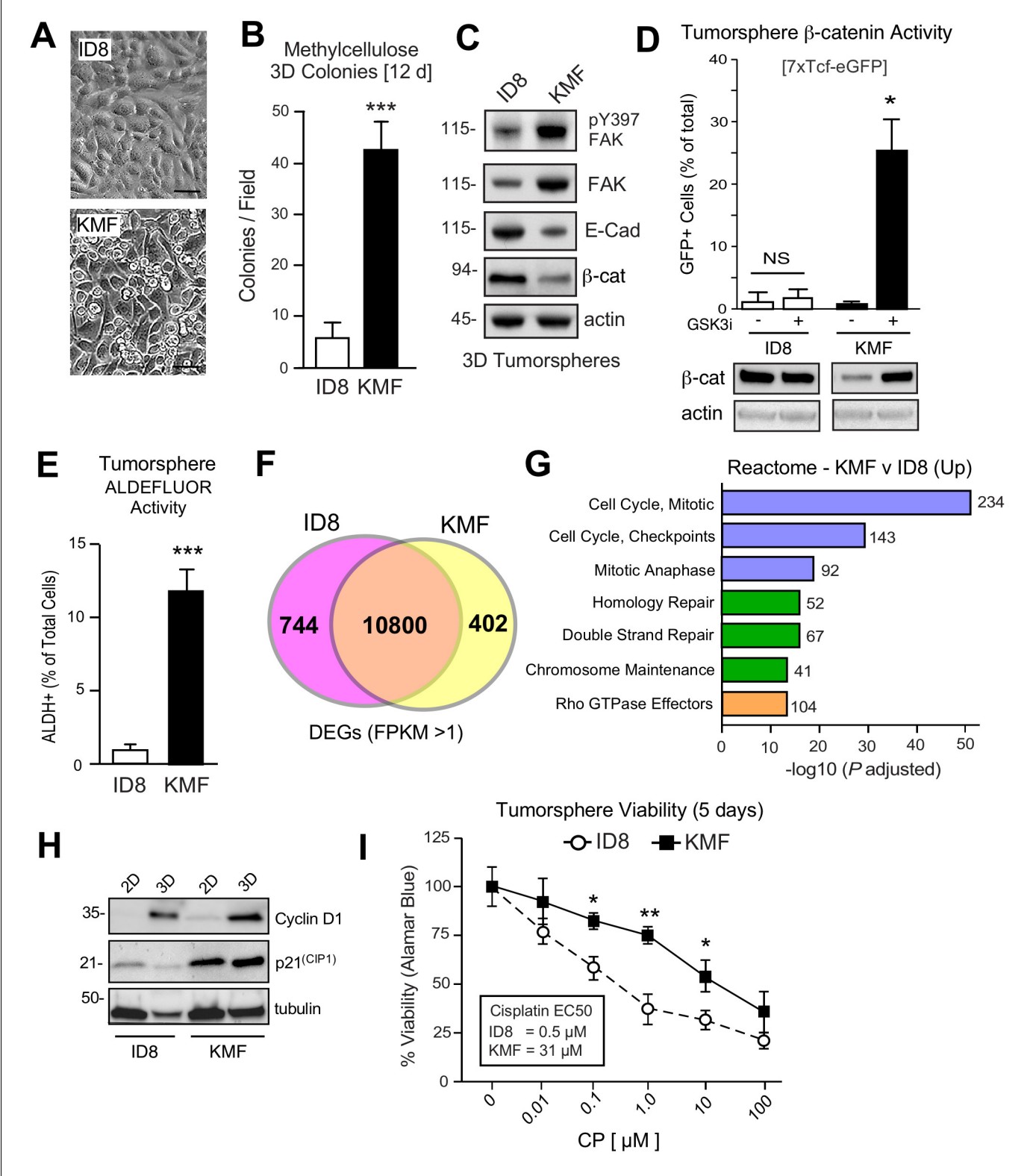

**Figure 2.** Acquired CSC phenotypes and greater intrinsic cisplatin resistance of KMF cells. (**A**) ID8 or KMF cells at high densities in 2D culture by phase-contrast imaging. Scale is 25 μm. (**B**) Quantitation of ID8 and KMF colony formation in methylcellulose (21 days). Values are means (± SEM, ***p<0.001, unpaired T-test) from three independent experiments. (**C**) ID8 and KMF 3D protein lysates immunoblotted for pY397 FAK, total FAK, E-cadherin, β-catenin, and actin. (**D**) Lentiviral-delivered β-catenin transcriptional reporter activity (7X TCF-eGFP) in ID8 and KMF cells grown as tumorspheres + /-
*Figure 2 continued on next page*

*Figure 2 continued*

GSK3β inhibitor. Values are percent GFP+ cells by flow cytometry (NS, not significant, *p<0.05, unpaired T-test, two experiments). Lower, lysates of cells immunoblotted for β-catenin and actin. (E) Quantitation of ID8 and KMF tumorsphere ALDEFLUOR activity. Values are means expressed as fold-change to ID8 (± SD, ***p<0.001, unpaired T-test, three independent experiments). (F) RNA sequencing Venn plot: number of shared or different expressed genes (DEGs) from ID8 and KMF cells in 3D culture. DEGs from FPKM (Fragments Per Kilobase of transcript per Million mapped read) values greater than 1. (G) Partial list of Reactome (top 20) KMF UP DEGs. N is the number of target genes elevated in KMF versus ID8. X axis are -log10 adjusted pP values. (H) Immunoblotting for cyclin D1, p21(Cip1), and tubulin in lysates of ID8 or KMF cells grown in 2D [10% serum] or 3D [serum-free PromoCell, 5 days] conditions. (I) Tumorsphere cytotoxicity (Alamar Blue) with increasing CP (5 days) expressed as percent viability to DMSO control. Means (n = 2) from four independent experiments (± SEM, *p<0.05, **p<0.01 by two-way ANOVA with a Bonferroni's multiple comparisons test). EC50 values independently determined.

DOI: https://doi.org/10.7554/eLife.47327.008

The following source data is available for figure 2:

**Source data 1.** Annotated RNA sequencing results from ID8 and ID8-IP/KMF tumorsphere cell lysates.

DOI: https://doi.org/10.7554/eLife.47327.009

(*Wiechert et al., 2016*). FAK protein and FAK tyrosine phosphorylation (pY397 FAK) levels are elevated in primary HGSOC tumors compared to normal tissue (*Zhang et al., 2016*), but it is not known whether chemotherapy alters FAK phosphorylation. To evaluate this, serial sections of paired primary biopsies and tumors obtained at the time of cytoreductive surgery following neoadjuvant carboplatin and paclitaxel chemotherapy were analyzed by immunohistochemical staining and quantitative image analyses (*Figure 3—figure supplement 1*).

A high degree of Pax8 (tumor marker) and pY397 FAK co-localized staining was detected in primary biopsy samples with FAK pY397 exhibiting both cytoplasmic and nuclear localization (*Figure 3A,B*). Several of these tumor cells stained positive for the Ki-67 proliferation marker. FAK pY397 staining was higher in ovarian tumor compared to surrounding stromal cells (*Figure 3—figure supplement 1*). Surprisingly, pY397 FAK staining remained elevated in non-necrotic tumor samples obtained after multiple cycles of neoadjuvant chemotherapy (*Figure 3—figure supplement 1*). By comparing samples from the same patients pre- and post-chemotherapy, we found pY397 FAK levels trended significantly upward in tumors surviving neoadjuvant chemotherapy (*Figure 3D*), further supporting an association between FAK signaling and tumor chemoresistance.

## FAK activation upon ovarian tumorsphere formation

FAK pY397 is canonically considered a marker associated with cell adhesion or increased tissue stiffness (*Sulzmaier et al., 2014*). Unexpectedly, pY397 FAK staining was also observed within Pax-8-positive ascites tumorspheres that also displayed active β-catenin staining (*Figure 4A*). This was unanticipated, since FAK Y397 phosphorylation is rapidly lost when human platinum-resistant OVCAR3 cells are removed from adherent 2D culture and placed in suspension (*Figure 4B*). However, extended time course analyses of OVCAR3 cells cultured in anchorage-independent PromoCell media revealed that FAK Y397 phosphorylation was restored as OVCAR3 cells clustered to form tumorspheres within 2–3 days (*Figure 4B,C*). Surprisingly, CP (1 µM) treatment of OVCAR3 tumorspheres (EC50 >10 µM) triggered increased FAK Y397 and β-catenin Y142 phosphorylation (*Figure 4D*). As β-catenin Y142 is a direct FAK substrate promoting β-catenin activation in endothelial cells (*Chen et al., 2012*), our findings support the notion that adhesion-independent non-canonical FAK activation occurs during tumorsphere formation and in response to CP stimulation.

## Combinatorial effects of CP and FAK inhibition

As increased FAK Y397 phosphorylation can occur upon CP treatment, we investigated the effect of low dose CP treatment (1 µM) in the presence or absence of a FAK inhibitor (VS-4718, 1 µM) over 5 days on tumorsphere formation, ALDH activity, and cell viability (*Figure 5*). Cisplatin EC50 values for growth inhibition were 13 µM and 31 µM for OVCAR3 and KMF tumorspheres, respectively. CP treatment resulted in increased tumorsphere formation and ALDEFLUOR activity in OVCAR3 and KMF cells, consistent with this low CP dose serving as an activation-type stress (*Figure 5A,B*). In contrast, FAK inhibitor (FAKi) reduced tumorsphere formation and ALDEFLUOR activity compared to control-treated OVCAR3 and KMF cells (*Figure 5A,B*).

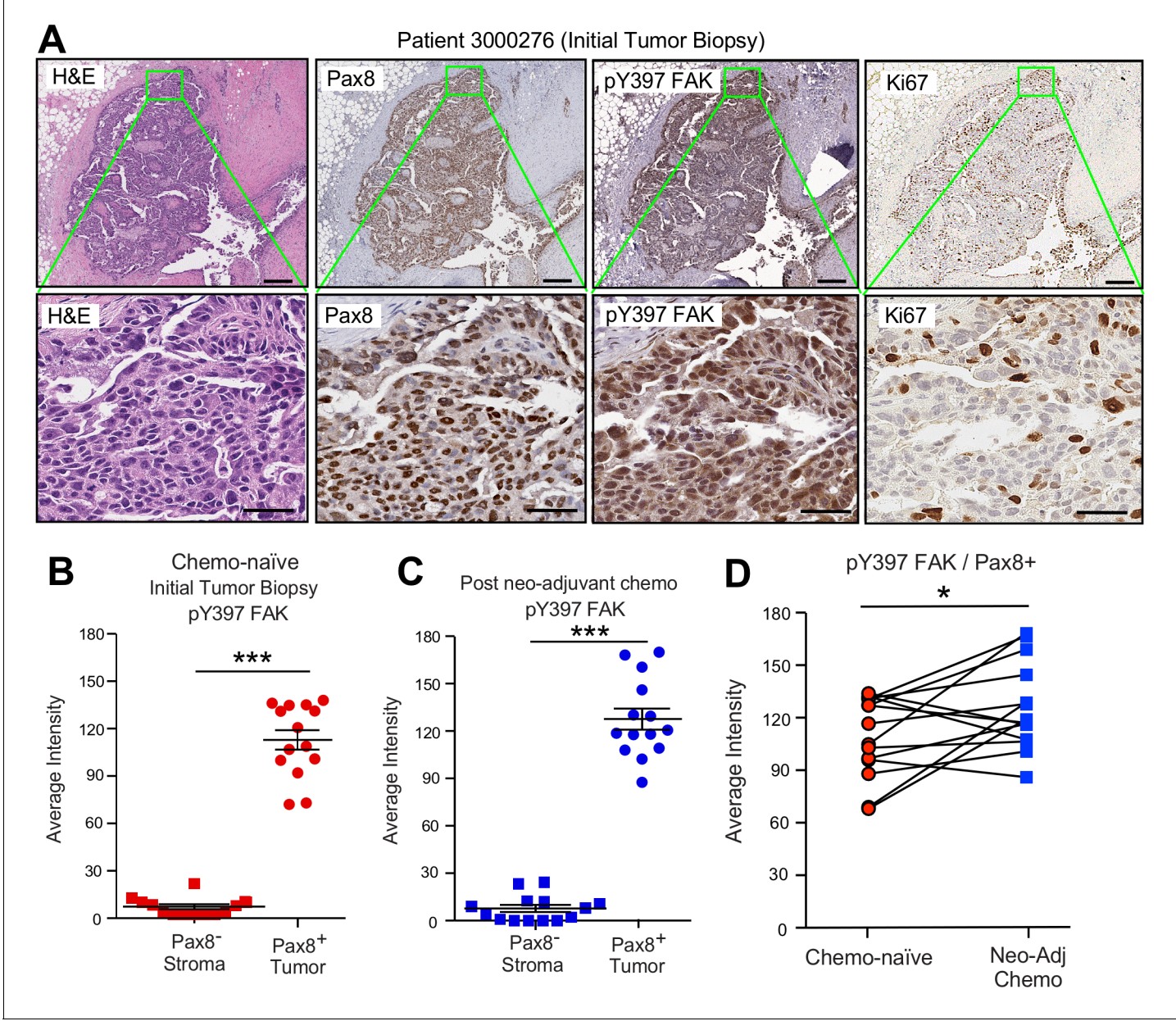

**Figure 3.** FAK Y397 phosphorylation (pY397) in HGSOC patient tumors surviving neoadjuvant chemotherapy. (**A**) IHC staining of paraffin-embedded serial initial tumor biopsy sections (patient 3000276) with H and E, Pax8, pY397 FAK, and Ki67. Scale is 200 µm. Inset (green box) region is shown at 40X (below). Scale is 60 µm. (**B and C**) FAK pY397 staining intensity of paired patient ovarian tumor samples from initial biopsies (panel B) and after surgical removal following neoadjuvant chemotherapy (panel C) within Pax8-positive (tumor) and Pax8-negative (stroma) regions. Dot plots are quantified staining from 14 paired patient samples (Aperio software) and bars show mean ± SEM (analyzed 11 regions per sample, ***p<0.001, unpaired T-test). (**D**) Increased FAK pY397 staining within Pax8-positive regions post-chemotherapy (*p<0.05, paired T-test). Lines are connecting paired patient tumor samples collected prior to and after neoadjuvant chemotherapy.

DOI: https://doi.org/10.7554/eLife.47327.010

The following figure supplements are available for figure 3:

**Figure supplement 1.** Patient tumor samples pre- and post-neoadjuvant chemotherapy, qualitative IHC score, and summary of quantitative image analyses.

DOI: https://doi.org/10.7554/eLife.47327.011

**Figure supplement 2.** FAK pY397 phosphorylation is maintained in Pax8-positive HGSOC tumors after neo-adjuvant chemotherapy.

DOI: https://doi.org/10.7554/eLife.47327.012

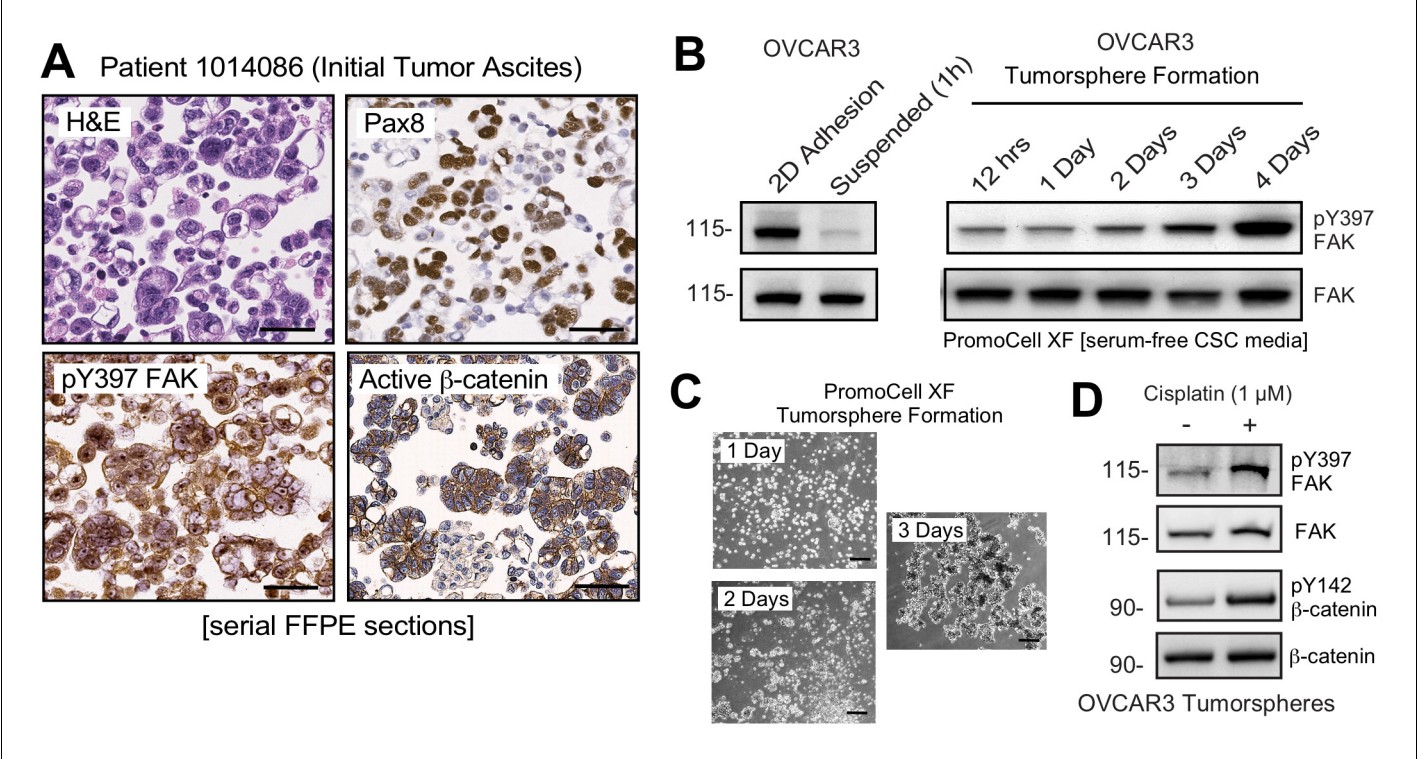

**Figure 4.** Non-canonical FAK Y397 phosphorylation in tumorspheres. (**A**) Paraffin-embedded IHC serial section staining (H and E, Pax8, pY397 FAK, and active β-catenin) of peritoneal ascites cells (tumorspheres) from initial (patient 1014086) biopsy. (**B**) OVCAR3 lysates from 2D adherent, suspended (1 hr), and cells in anchorage-independent serum-free (PromoCell) conditions facilitating tumorsphere formation were analyzed by total FAK and pY397 FAK immunoblotting. (**C**) Representative images of OVCAR3 tumorsphere formation at Day 1, Day 2, and Day 3. Scale is 2 mm. (**D**) OVCAR3 cells as tumorspheres (Day 3) treated with DMSO or CP (1 µM) for 1 hr and protein lysates blotted for pY397 FAK, total FAK, pY142 β-catenin, and total β-catenin.

DOI: https://doi.org/10.7554/eLife.47327.013

FAKi was not directly cytotoxic, since only CP combined with FAKi decreased tumorsphere viability (*Figure 5C*). Single agent CP or FAKi treatment did not alter KMF (*Figure 5—figure supplement 1*) or OVCAR3 (*Figure 5—figure supplement 2*) growth or viability in 2D culture. Under 3D conditions, FAKi reduced FAK Y397 phosphorylation and resulted in an elevated percentage of KMF and OVCAR3 cells in G1 phase of the cell cycle (*Figure 5—figure supplements 1* and *2*). The finding that FAKi decreased 3D cell proliferation, and that FAKi exhibits combinatorial activity with low-dose CP to promote apoptosis, highlights a potential therapeutic combination.

## Platinum-resistant tumorspheres can acquire dependence on FAK for growth

Phosphoinositide 3-kinase (PI3K)-elicited Akt activation is one of several survival signaling pathways downstream of FAK (*Sulzmaier et al., 2014*). More than half of HGSOC tumors harbor genetic lesions that can elevate PI3K activity (*Hanrahan et al., 2012*). A2780 human ovarian carcinoma tumor cells contain activating mutations in *PI3KCA* and inactivation of *PTEN* - alterations that can promote Akt activation (*Domcke et al., 2013*). OVCAR10 cells similarly exhibit elevated Akt phosphorylation and both A2780 and OVCAR10 cells are resistant to FAKi (1 µM) effects on 3D cell proliferation (*Tancioni et al., 2014*). To determine if in vitro acquisition of increased CP resistance alters FAK signaling, intermittent CP exposure (10 µM) and cell recovery was used to generate OVCAR10-CP (EC50 = 9 µM) and maintain A2780-CP70 cell (EC50 = 60 µM) selection. Immunoblotting revealed constitutively elevated FAK pY397 within tumorspheres of CP-resistant compared to parental cells (*Figure 6A*). In addition, we find that CP-resistant A2780-CP70 and OVCAR10-CP cells exhibited a

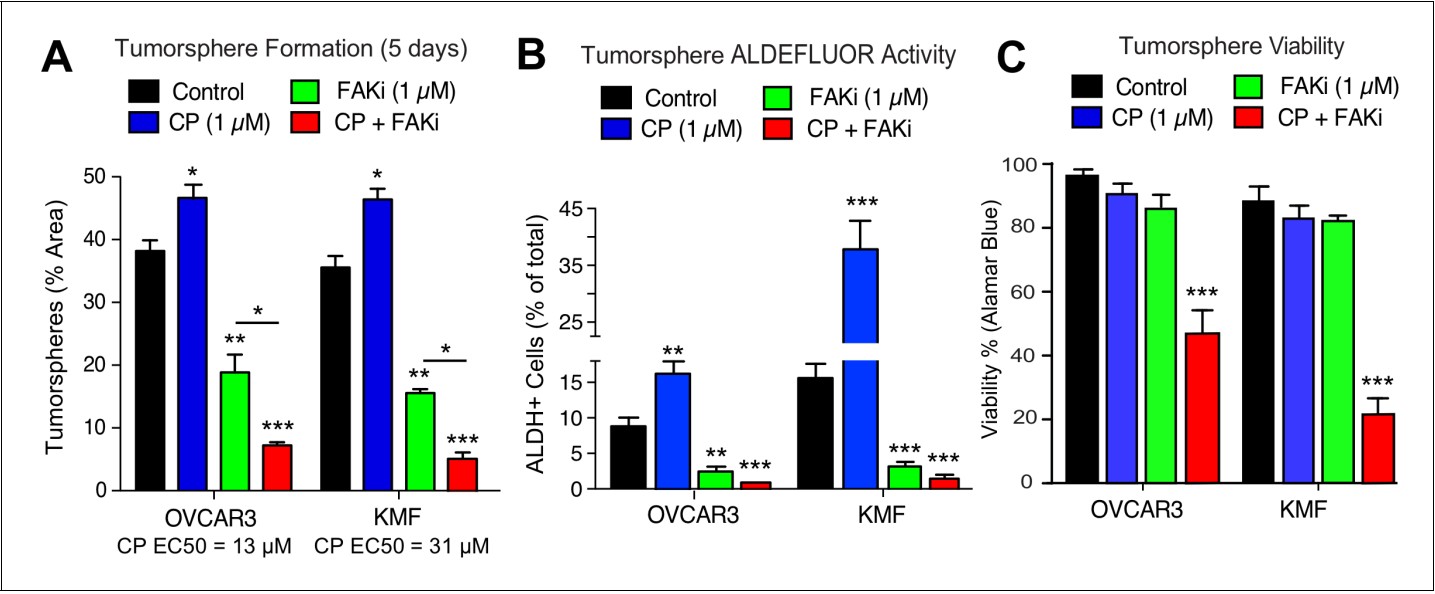

**Figure 5.** Prevention of CSC phenotypes in vitro by pharmacological FAK inhibition. Quantification of OVCAR3 and KMF tumorsphere formation (panel A), ALDEFLUOR activity (panel B), and tumorsphere viability (panel C) in the presence of DMSO (control), CP (1 μM), FAKi (VS-4718, 1 μM) or CP plus FAKi for 5 days. Values are means (± SEM, *p<0.05, **p<0.01, ***p<0.001 unpaired T-test) of three independent experiments. Panel C, values are means (± SEM, ***p<0.001, one-way ANOVA) from three independent experiments.

DOI: https://doi.org/10.7554/eLife.47327.014

The following figure supplements are available for figure 5:

**Figure supplement 1.** Small molecule FAK inhibition prevents KMF 3D tumorsphere proliferation with effects on cell cycle but not cell apoptosis.
DOI: https://doi.org/10.7554/eLife.47327.015

**Figure supplement 2.** Small molecule FAK inhibition selectively inhibits OVCAR3 3D tumorsphere proliferation with effects on cell cycle but not cell apoptosis.
DOI: https://doi.org/10.7554/eLife.47327.016

newly acquired dose-dependent sensitivity to FAKi growth inhibition as tumorspheres (*Figure 6B*), but not when the same cells were grown in 2D conditions (*Figure 6B—figure supplement 1*).

FAKi treatment of A2780-CP70 and OVCAR10-CP tumorspheres was accompanied by an increased percentage of G1 phase cells, decreased cyclin D1 expression, but not increased apoptosis (*Figure 6—figure supplement 2*). Both A2780-CP70 and OVCAR10-CP tumorspheres possessed increased ALDH activity compared to parental cells (*Figure 6—figure supplement 3*) and this was dependent on FAK activity (*Figure 6E*). FAKi selectively prevented A2780-CP70 and OVCAR10-CP methylcellulose colony formation (*Figure 6F*) but did not inhibit parental A2780 or OVCAR10 colony formation (*Figure 6—figure supplement 2*).

To determine the effects of combinatorial FAKi and low dose CP treatments, OVCAR10-CP colony formation was evaluated in the presence of DMSO (control), FAKi (1 μM), CP (1 μM), or FAKi and CP combination (*Figure 6G,I*). Single agent FAKi reduced colony size (*Figure 6G*), consistent with an inhibitory effect on tumorsphere proliferation. FAKi with CP prevented colony formation (*Figure 6H*) and this was associated with increased OVCAR10-CP apoptosis (*Figure 6I*). Only the combination of FAKi with CP triggered increased A2780-CP70 apoptosis (*Figure 6I*). These results support the notion that selection for platinum resistance can result in the acquired dependence on FAK activity for tumorsphere growth. Moreover, FAK inhibition in combination with CP can trigger CP-resistant tumorsphere apoptosis.

## FAK inhibition sensitizes CP-resistant A2780-CP70 tumors to chemotherapy

DTomato plus luciferase-labeled A2780 or A2780-CP70 cells were orthotopically injected into mice to assess the combinatorial potential of FAKi (VS-4718) and cisplatin plus paclitaxel (CPT) chemotherapy on paired CP-sensitive and CP-resistant tumors established in immune-deficient mice

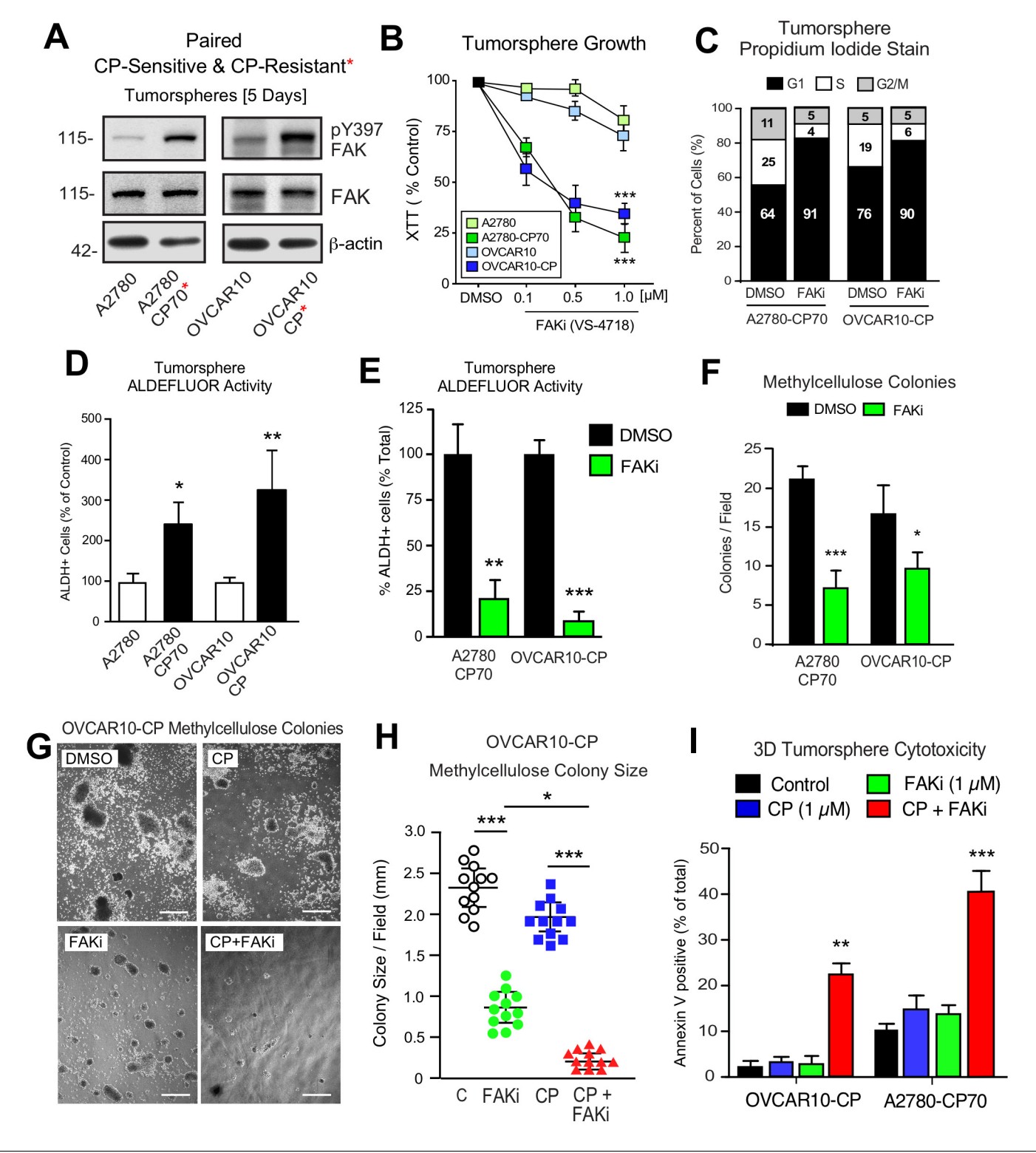

**Figure 6.** Acquired FAK dependence for CP-resistant tumorsphere growth. (**A**) Human A2780, A2780-CP70, OVCAR10, and OVCAR10-CP tumorsphere lysates immunoblotted for FAK pY397, total FAK, and actin. (**B**) Growth of A2780, OVCAR10, A2780-CP70, or OVCAR10-CP cells as tumorspheres in the presence of FAKi (VS-4718, 0.1 to 1.0 µM) for 4 days. Values are means (± SEM, ***p<0.001, one-way ANOVA) from two independent experiments. (**C**) A2780-CP70 or OVCAR10-CP cells grown as tumorspheres (3 days) were treated with DMSO or FAKi (VS-4718, 1 µM) for 24 hr, stained with propidium iodide, and analyzed by flow cytometry. Shown is percent of cells in G0/G1, S, or G2/M phase of the cell cycle. (**D**) Quantitation of A2780-CP70 and

*Figure 6 continued on next page*

*Figure 6 continued*

OVCAR10-CP colony formation in methylcellulose (21 days) with DMSO (control) or FAKi (VS-4718, 1 μM). Values are means (± SEM, *p<0.05, ***p<0.001, unpaired T-test) from two independent experiments. (E) A2780-CP70 and OVCAR10-CP tumorsphere ALDEFLUOR activity treated with DMSO or FAKi (VS-4718, 1 μM) for 24 hr. Values are means (± SEM, **p<0.01, ***p<0.001, one-way ANOVA compared to DMSO) for three independent experiments. (F) Quantitation of A2780-CP70 and OVCAR10-CP methylcellulose colony formation (21 days). Values are means (± SEM, *p<0.05, ***p<0.001, unpaired T-test) from two independent experiments. (G and H) Representative OVCAR10-CP methylcellulose colony formation (21 days) (panel G) and colony size (panel H) in the presence of DMSO (control), CP (1 μM), FAKi (1 μM), or CP plus FAKi. Scale is 2.5 mm. Values are means (± SEM, *p<0.05, ***p<0.001, one-way ANOVA) from two independent experiments. (I) A2780-CP70 and OVCAR10-CP tumorsphere cytotoxicity (annexin V) in the presence of DMSO (control), CP (1 μM), FAKi (1 μM), or CP plus FAKi. Values are means (± SEM, **p<0.01, one-way ANOVA) from three independent experiments.

DOI: https://doi.org/10.7554/eLife.47327.017

The following figure supplements are available for figure 6:

**Figure supplement 1.** Constitutively elevated FAK Y397 phosphorylation in CP-resistant tumorspheres.

DOI: https://doi.org/10.7554/eLife.47327.018

**Figure supplement 2.** Platinum-resistant A2780-CP70 exhibit FAK-dependent growth.

DOI: https://doi.org/10.7554/eLife.47327.019

**Figure supplement 3.** ALDEFLUOR assays.

DOI: https://doi.org/10.7554/eLife.47327.020

(*Figure 7*). A2780 tumor growth was insensitive to FAKi (*Figure 7—figure supplement 1*), consistent with limited FAKi effects on A2780 growth in vitro. As expected, CPT chemotherapy inhibited A2780 tumor growth, but did not affect the resistant A2780-CP70 tumors. FAKi with CPT did not yield additional anti-tumor effects on A2780 tumor growth. In dramatic contrast, in A2780-CP70 tumors, single-agent FAKi treatment reduced tumor mass approximately 40% compared to controls (*Figure 7A,D*), and the combination of FAKi with CPT chemotherapy resulted in an even greater reduction in tumor growth (*Figure 7B–D*). Interestingly, CPT treatment increased FAK Y397 phosphorylation (*Figure 7E,F*) and ALDH-1A1 staining (*Figure 7G*) in non-necrotic regions of A2780-CP70 tumors (*Figure 7—figure supplement 2*). Adding FAKi to CPT chemotherapy suppressed FAK Y397 phosphorylation, reduced ALDH-1A1 tumor staining, and increased tumor apoptosis in vivo (*Figure 7E,H*). These results show that the combination of a FAK inhibitor with CP exhibits selective anti-tumor effects on CP-resistant A2780-CP70 tumors in vivo.

## KMF FAK inactivation and reconstitution link intrinsic FAK activity to β-catenin

To provide genetic support for the role of FAK in intrinsic CP resistance, CRISPR/Cas9 targeting was used to inactivate murine *Ptk2* exon four in KMF cells (*Figure 8*) and human *PTK2* exon 3 of OVCAR3 cells (*Figure 8—figure supplement 1*) to create FAK knockout (KO) cells. No difference in 2D adherent cell growth was detected and FAK KO clones were isolated. Sanger sequencing confirmed unique deletions/insertions predicted to terminate FAK protein translation in each of four *Ptk2* alleles identified in KT3 and KT13 FAK KO clones (*Figure 8—figure supplement 2*). Exome sequencing of FAK KO clone KT13 (90% of exons sequenced at 100X) detected only 165 unique variants, including the four *Ptk2* alterations, indicating that CRISPR targeting was specific and that the FAK KO KT13 genome is similar to parental KMF cells (*Figure 8—source data 1*).

CRISPR inactivated FAK but not expression of the FAK-related Pyk2 kinase (*Figure 8A*). In KT3 and KT13 clones, total β-catenin protein levels were constitutively lower than KMF cells (*Figure 8A*) and this corresponded to decreased β-catenin transcriptional activity (*Figure 8B*). When cultured in PromoCell tumorsphere media under anchorage-independent conditions, FAK KO cell viability remained high after 5 days (*Figure 8C*). However, FAK KO cells exhibited sensitivity to CP-mediated cytotoxicity compared to parental KMF cells. Similar results were obtained comparing parental and FAK KO OVCAR3 cells (*Figure 8—figure supplement 1*). Importantly, phenotypes of decreased tumorsphere formation, ALDEFLUOR activity, and CP resistance in FAK KO OVCAR3 cells were rescued by GFP-FAK re-expression (*Figure 8—figure supplement 1*). These results connect FAK expression to CP resistance and CSC-associated phenotypes.

To establish a genetic linkage with FAK activity, FAK KO KT13 cells were stably reconstituted with GFP-FAK wildtype (WT) or a catalytically-inactive (K454R) GFP-FAK point mutation (*Figure 8D*)

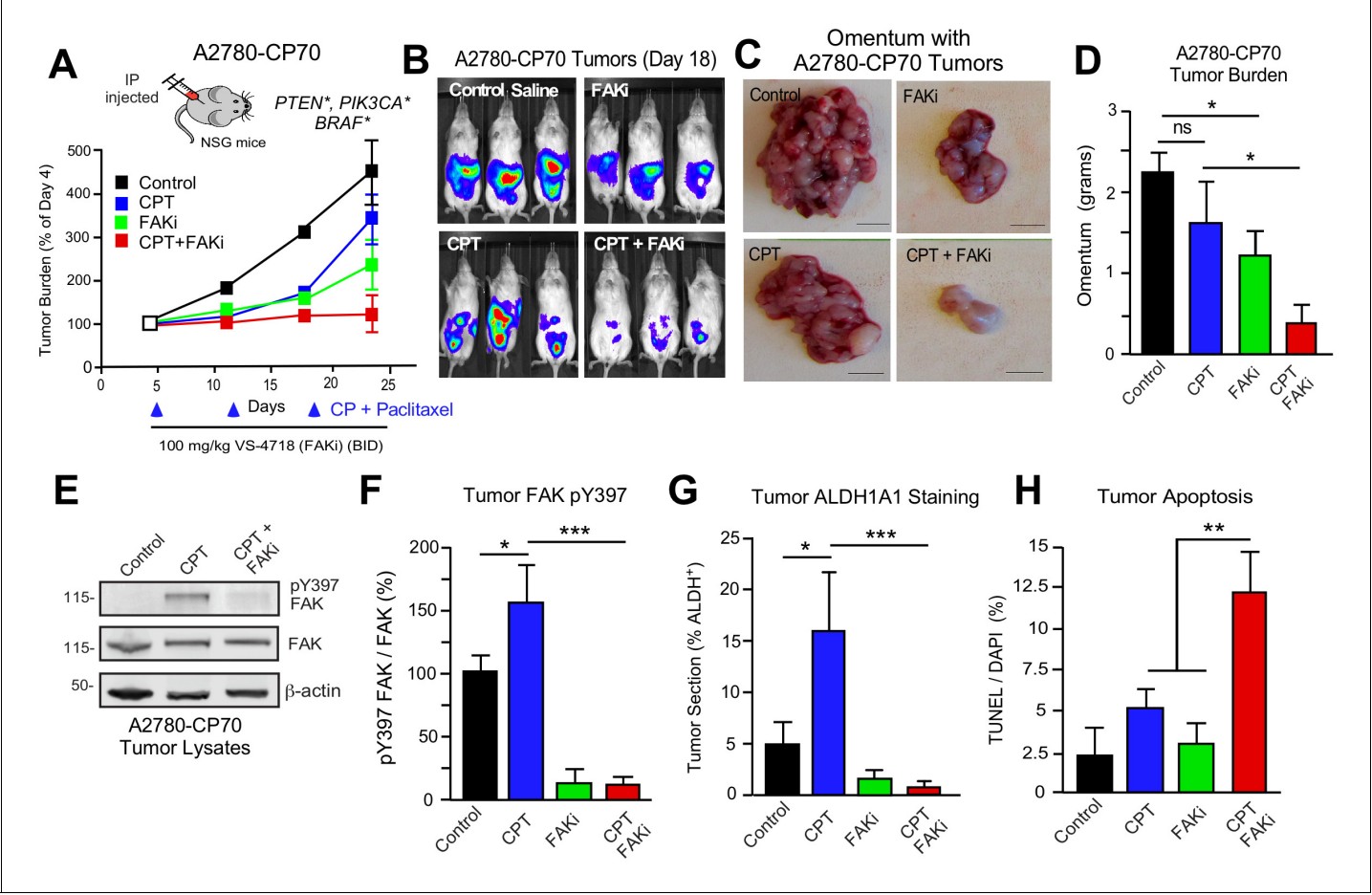

**Figure 7.** FAK inhibition sensitizes CP-resistant tumors chemotherapy-induced apoptosis. (**A**) Experimental schematic and IVIS imaging of labeled A2780-CP70 cells IP injected into NSG mice (randomized at Day 5). Experimental groups: control saline (black) injection on Days 5, 12, and 19; VS-4718 by oral gavage (green, 100 mg/kg, BID); CPT chemotherapy injection (blue, 3 mg/kg cisplatin and 2 mg/kg paclitaxel) on Days 5, 12, and 19; and VS-4718 plus CPT combined administration (red). IVIS imaging was performed on Days 4, 11, 18, and 23. Tumor burden is expressed as percent of Day 4. (**B**) Representative IVIS images of A2780-CP70 tumor burden on Day 18. (**C**) Representative images of omentum with A2780-CP70 tumors at Day 24. Scale is 0.5 cm. (**D**) Omentum-associated A2780-CP70 tumor mass (n = 6,± SEM *p<0.05, one-way ANOVA) from each treatment group. (**E**) A2780-CP70 tumor lysates immunoblotted for FAK pY397, FAK, and actin. (**F**) Ratio of pY397 FAK to total FAK levels in tumor lysates by immunoblotting. Values are means (± SEM *p<0.05, ***p<0.001, one-way ANOVA) of three tumors per experimental group. Control set to 100. (**G** and **H**) Percent ALDH-1A1-positive immunofluorescent A2780-CP70 tumor staining or apoptosis (TUNEL and Hoescht 33342 staining) in A2780-CP70 tumors. Values are means (± SEM, two independent tumors, five random fields per tumor at 20X, *p<0.05, **p<0.01, ***p<0.001 one-way ANOVA).

DOI: https://doi.org/10.7554/eLife.47327.021

The following figure supplements are available for figure 7:

**Figure supplement 1.** Inhibition of A2780 tumor growth by cisplatin-paclitaxel (CPT) chemotherapy.
DOI: https://doi.org/10.7554/eLife.47327.022
**Figure supplement 2.** Elevated FAK Y397 phosphorylation and ALDH staining in non-necrotic regions of CPT-treated mice with A2780-CP70 tumors.
DOI: https://doi.org/10.7554/eLife.47327.023

(*Lim et al., 2010*). In 3D anchorage-independent conditions, GFP-FAK-WT and GFP-FAK-R454 were equally expressed, but only GFP-FAK-WT was phosphorylated at Y397 (*Figure 8D*). This result confirms that intrinsic FAK kinase activity facilitates FAK Y397 phosphorylation. To identify proteomic differences in an unbiased manner, lysates of KT13 FAK KO, FAK-WT, and FAK-R454 cells were analyzed by mass spectrometry (*Figure 8—source data 2*). Elevated levels of extracellular matrix (collagen type six and laminin), surface receptors (N-cadherin and Nectin-2), β-catenin, and Wnt signaling targets (GPC4) (*Sakane et al., 2012*) were present in FAK-WT compared to FAK KO cells. These

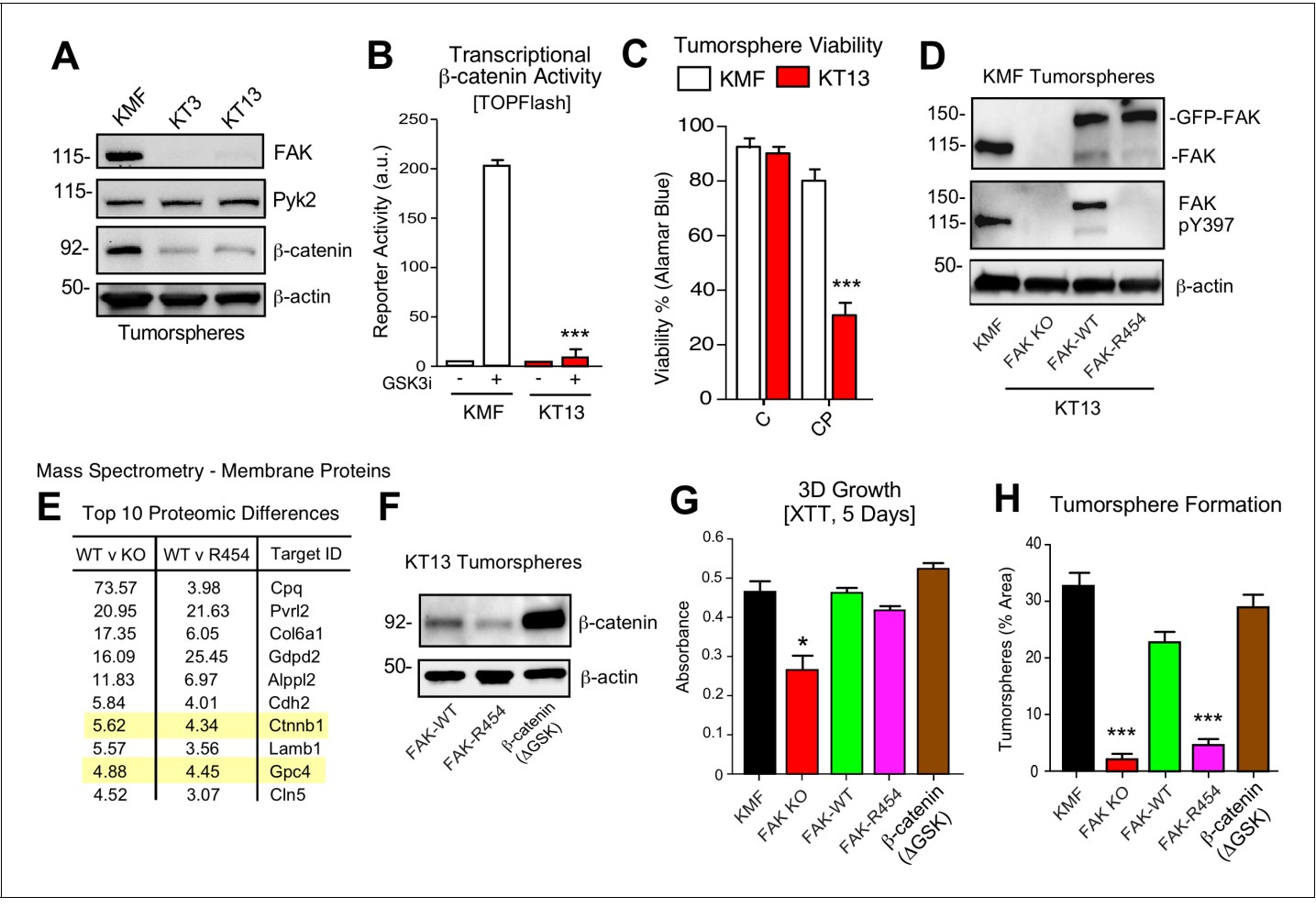

**Figure 8.** KMF FAK KO and re-expression link intrinsic FAK activity to β-catenin and tumorsphere formation. (**A**) Immunoblotting of KMF and CRISPR-mediated FAK KO clones KT3 and KT13 cell lysates for FAK, Pyk2, β-catenin, and actin. (**B**) KMF and FAK KO KT13 cell viability treated with DMSO (control) or CP (1 μM) after 72 hr as measured by Alamar Blue. Values are means (± SEM, *p<0.05, **p<0.01, ***p<0.001, one-way ANOVA with Fisher's LSD multiple comparison test) for three independent experiments. (**C**) β-catenin transcriptional reporter activity (TOPFlash) in transfected KMF and KT13 FAK KO cells + /- GSK3β inhibitor. Values are arbitrary units (***p<0.001, unpaired T-test, two independent experiments). (**D**) Immunoblotting for pY397 FAK, FAK, and actin in lysates of KMF, FAK KO, GFP-FAK-WT, and GFP-FAK-R454 re-expressing cells. (**E**) Top 10 proteomic differences (fold-change) detected by mass spectroscopy of membrane associated proteins in KT13 FAK KO, GFP-FAK-WT, and GFP-FAK R454 re-expressing cells. (**F**) Immunoblotting for β-catenin and actin in lysates of KT13 FAK KO cells stably expressing GFP-FAK-WT, GFP-FAK-R454, or β-catenin (ΔGSK). (**G**and **H**) XTT metabolic activity (panel G) or tumorsphere formation (panel H) of KMF, KT13 FAK KO, or the indicated reconstituted cells in PromoCell after 5 days. Values are means (± SEM, *p<0.05, ***p<0.001, one-way ANOVA with a Tukey's multiple comparisons test) from 2 (panel G) or 3 (panel H) independent experiments.

DOI: https://doi.org/10.7554/eLife.47327.024

The following source data and figure supplements are available for figure 8:

**Source data 1.** KMF FAK KO clone KT13 exome sequencing variants.
DOI: https://doi.org/10.7554/eLife.47327.027
**Source data 2.** Summary of mass spectrometry-detected proteomic changes between KMF FAK KO, FAK-WT, and FAK kinase-inactive (K454R) re-expressing cells grown as tumorspheres.
DOI: https://doi.org/10.7554/eLife.47327.028
**Figure supplement 1.** OVCAR3 CRISPR-mediated FAK KO and re-expression.
DOI: https://doi.org/10.7554/eLife.47327.025
**Figure supplement 2.** Sequencing validation of CRISPR/Cas9-mediated FAK KO in KMF cells.
DOI: https://doi.org/10.7554/eLife.47327.026

differences were maintained in FAK-WT versus FAK-R454 cells (*Figure 8E*). As FAK can regulate β-catenin levels in colon carcinoma cells (*Gao et al., 2015*), the mass spectrometry results implicate intrinsic FAK activity in supporting Wnt-β-catenin signaling in KMF cells.

## β-catenin promotes FAK KO tumorsphere formation, ALDEFLUOR activity, and CP resistance

To test whether stabilized β-catenin was sufficient to complement KMF FAK KO cell phenotypes, an activated β-catenin point-mutant (ΔGSK) lacking the regulatory GSK3β phosphorylation sites (*Barth et al., 1999*) was expressed in KT13 FAK KO cells (*Figure 8F*). A series of assays were performed comparing KMF, FAK KO, FAK-WT, FAK-R454, and FAK KO β-catenin ΔGSK expressing cells. In 3D conditions, FAK KO proliferation was less than KMF cells and this was rescued by FAK-WT, FAK-R454, and β-catenin ΔGSK (*Figure 8G*). Notably, FAK-R454 cells grew in 3D culture, whereas parental KMF cells treated with FAKi exhibit growth defects (*Figure 5*). In contrast, FAK activity was required for clustering of KMF cells into tumorspheres and this phenotype was also supported by β-catenin ΔGSK expression (*Figure 8H*). FAK-WT restored total ALDEFLUOR activity, ALDH-1A2, ALDH-1B1, and Myc protein levels in FAK KO cells equivalent to parental KMF cells (*Figure 9A,C*). Expression of FAK-WT and β-catenin ΔGSK but not FAK-R454 significantly enhanced FAK KO resistance to CP cytotoxicity in vitro (*Figure 9C*). Together, these results link intrinsic FAK activity and β-catenin in supporting KMF CSC and intrinsic CP resistance phenotypes.

## FAK activity is essential for KMF tumor growth

Although oral FAKi administration can inhibit tumor growth in mice (*Sulzmaier et al., 2014*), it remains unclear whether this is mediated by FAK inhibition within tumor, stroma, or multiple cell types. Parental KMF, FAK KO, and FAK-WT cells were labeled with a dual reporter (luciferase and dTomato) and injected within the intraperitoneal cavity of C57Bl/6 mice to test whether FAK is essential for tumor establishment. At Day 24, luciferase imaging revealed significant KMF tumor burden whereas FAK KO tumor cells were only weakly detected (*Figure 9D—figure supplement 1*). At Day 28, flow cytometry enumeration of dTomato-positive peritoneal cells revealed significantly fewer FAK KO compared KMF and FAK-WT tumor cells (*Figure 9E*). In an independent experiment over 21 days, FAK KO and FAK-R454 cells did not grow in vivo as did FAK-WT tumors (*Figure 9F*). Surprisingly, β-catenin ΔGSK also did not promote FAK KO tumor growth (*Figure 9F*). This result contrasted with the rescue of FAK KO tumorsphere formation, ALDEFLUOR activity, and CP resistance in vitro by β-catenin ΔGSK expression (*Figure 8*). Importantly, these results reveal that intrinsic FAK activity is essential for KMF tumor establishment in mice. Moreover, β-catenin signaling was not sufficient to support KMF tumor growth in the absence of FAK.

## Transcriptomic analyses identify common FAK activity-dependent and β-catenin supported mRNA targets in KMF and HGSOC

FAK controls various gene transcriptional networks (*Sulzmaier et al., 2014*; *Serrels et al., 2017*). As FAK KO cells are deficient in a number of different phenotypes, we performed RNA sequencing from KT13 FAK KO, FAK-WT, FAK-R454, and β-catenin ΔGSK cells grown in PromoCell to determine FAK activity-dependent, -independent, and β-catenin-specific patterns of differential gene expression. Using an FPKM cutoff greater than one, 1040 mRNA transcripts were increased two-fold or more by FAK compared to FAK KO cells and significant after multiple testing correction (*Figure 10—source data 1*). By filtering out transcripts that were elevated in FAK-R454 versus FAK KO cells (FAK activity-independent targets), 591 genes were identified as FAK activity-dependent and showed KEGG (Kyoto Encyclopedia of Genes and Genomes) pathway enrichment for *MAPK Signaling, Cell Cycle, Axon Guidance, and DNA Replication.*

After unbiased filtering of the 591 FAK activity-dependent transcripts with those elevated by β-catenin ΔGSK, 241 shared transcripts were identified as co-regulated by FAK activity and β-catenin (*Figure 10A*). KEGG enrichments were *MAPK Signaling, Cell Cycle, Hippo Signaling, and Pluripotency.* Last, the 241 shared FAK and β-catenin murine targets were filtered against genes elevated in HGSOC by querying TCGA. Notably, 135 targets matched, 77 exhibit frequent gains in 20% of tumors, and 19 genes were elevated in greater than 50% of HGSOC patient tumors (*Figure 10A*). Myc was the most common gene target and immunoblotting of KMF, FAK KO, and FAK-WT lysates

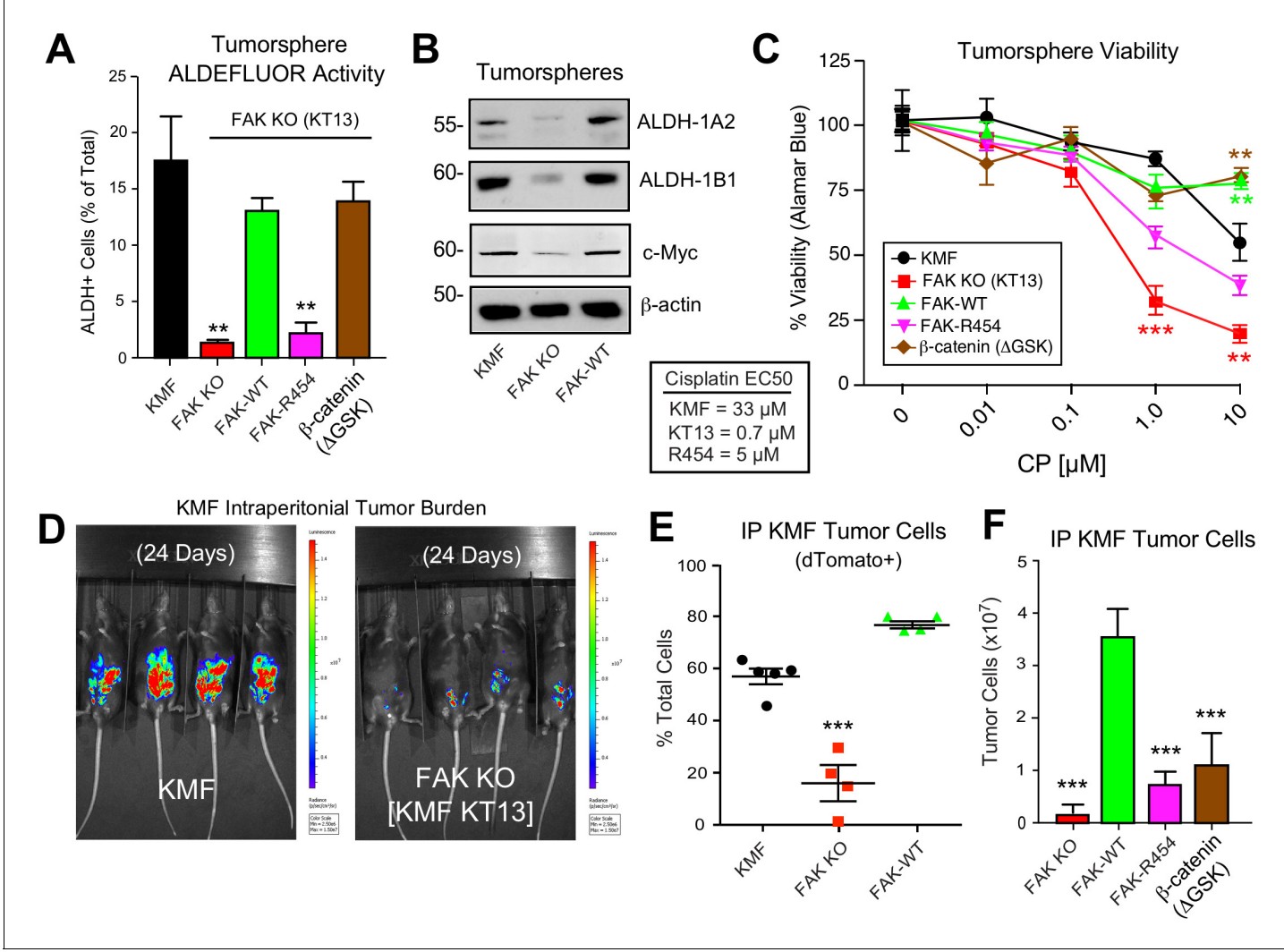

**Figure 9.** Intrinsic FAK activity supports ALDFLUOR activity, CP resistance, and is essential for KMF tumor growth. (**A**) ALDEFLUOR activity of KMF, KT13 FAK KO, or the indicated reconstituted cells in PromoCell after 5 days. Values are means (± SEM **p<0.01, ***p<0.001, one-way ANOVA with a Tukey's multiple comparisons test) of four independent experiments. (**B**) Immunoblotting for ALDH-1A2, ALDH-1B1, Myc, and actin in the indicated cell lysates. (**C**) Viability (Alamar Blue) of KMF (black circles), KT13 FAK KO (red squares), GFP-FAK WT (green triangle), GFP-FAK R454 (magenta triangle), and β-catenin ΔGSK (brown diamond) expressing cells treated with increasing CP concentrations for 5 days. Values are means (± SEM, **, p<0.01, ***p<0.001, two-way ANOVA with a Bonferroni's multiple comparisons test) from three independent experiments. Lower, EC50 values were determined independently and using Prism. (**D**) IVIS imaging of C57Bl/6 mice with dTomato+ and luciferase-expressing KMF or KT13 FAK KO cells at experimental Day 24. (**E**) Flow cytometry analyses of peritoneal wash collected dTomato+ cells at Day 28 of mice bearing KMF (black), KT13 FAK KO (red), and FAK KO re-expressing FAK WT (green) cells. Values are means expressed as percent of total cells in peritoneal wash (± SEM, ***p<0.001, one-way ANOVA). (**F**) Intraperitoneal (IP) tumor growth of KT13 FAK KO (red), GFP-FAK-WT (green), GFP-FAK-R454 (magenta), or β-catenin ΔGSK (brown) expressing cells. Values are means of CD45-negative tumor cells determined by flow cytometry (± SD, ***p<0.001, one-way ANOVA).
DOI: https://doi.org/10.7554/eLife.47327.029

The following figure supplement is available for figure 9:

**Figure supplement 1.** Comparison of KMF, KT13 FAK KO, and FAK KO re-expressing GFP-FAK-WT orthotopic growth in C57Bl/6 mice.
DOI: https://doi.org/10.7554/eLife.47327.030

confirmed the regulation of Myc protein expression by FAK (*Figure 9B*). Although Myc is a common target of β-catenin (*Sanchez-Vega et al., 2018*), this result is surprising as both *Myc* and *Ptk2* (FAK) DNA loci exhibit gains in KMF cells (*Table 1*). However, FAK knockout and FAK re-expression in human OVCAR3 cells also showed FAK-mediated regulation of Myc protein levels (*Figure 8—figure*

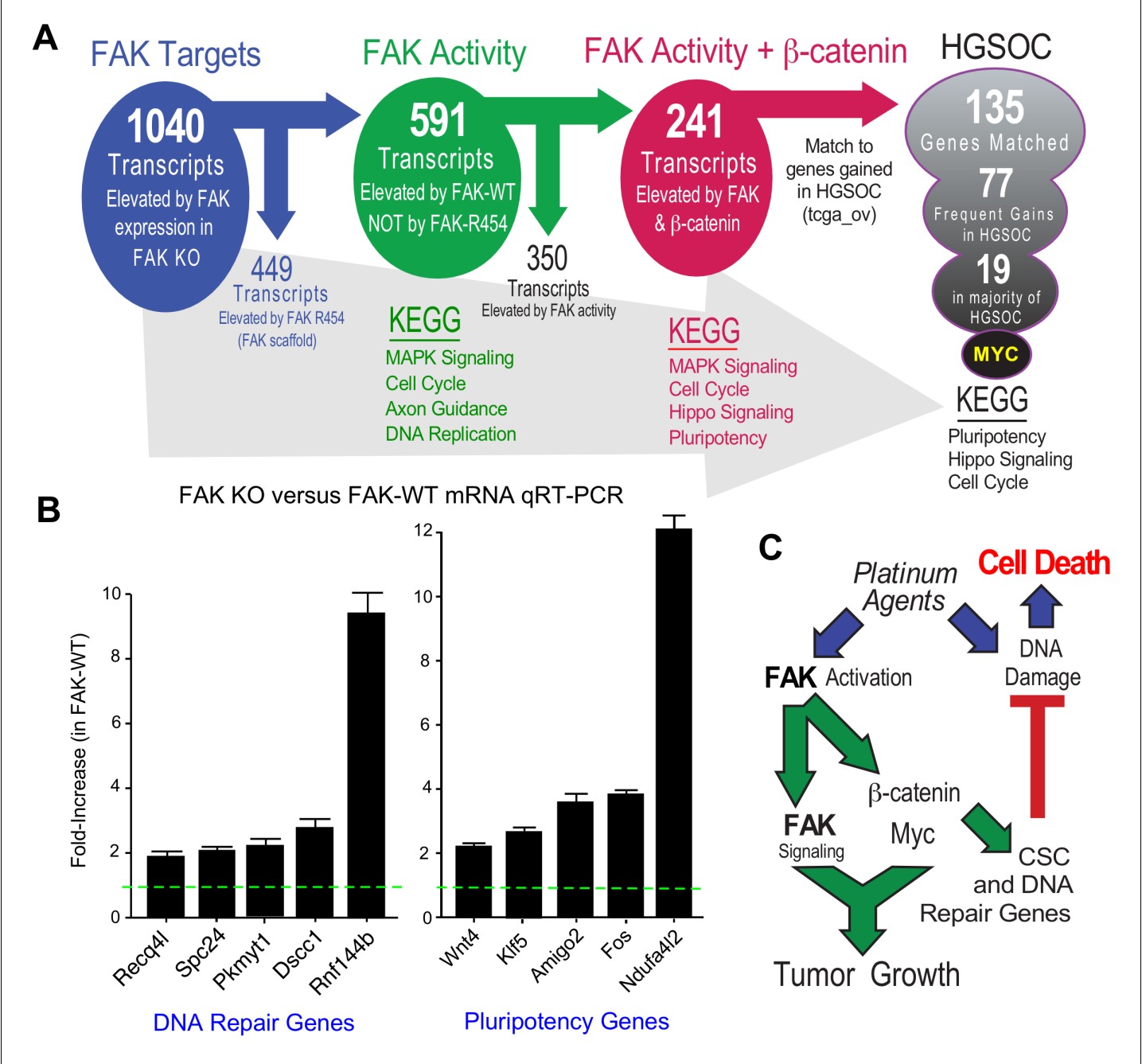

**Figure 10.** FAK activity and β-catenin promote a common gene signature elevated in HGSOC. (**A**) Summary of KMF RNA sequencing and filtering of differential gene expression. 1040 mRNAs were elevated (greater than log2 and FPKM >1) in FAK-WT versus KT13 FAK KO cells. 449 mRNAs were elevated in FAK R454 versus KT13 FAK KO cells. This represents FAK scaffold or activity-independent group (blue). By subtraction of FAK-R454 from FAK-WT targets, 591 FAK activity-dependent targets were identified (green). 1739 mRNAs were elevated in β-catenin ΔGSK cells, and by filtering against FAK activity-induced mRNAs, 241 common FAK activity and β-catenin enhanced mRNA targets were identified (red). 135 of 241 murine KMF targets were matched to genes elevated in HGSOC. 77 targets were elevated in 20% of HGSOC patients and 19 targets were elevated in more than 50% of HGSOC patients. *MYC* exhibits the highest genetic gain frequency. Top Kyoto Encyclopedia of Genes and Genomes (KEGG) pathway enrichments are listed for filtered groups. (**B**) Real-time PCR quantitation (qRT-PCR) of the indicated DNA repair- or pluripotency-associated mRNAs from FAK KO and FAK-WT cells grown in PromoCell for 5 days. Values were normalized to ribosomal RPL19 and fold increase are means from two replicates (± SEM, *p<0.05, T test). (**C**) Signaling summary of death-inducing and paradoxical survival-sustaining FAK activation by platinum chemotherapy. FAK signaling to β-catenin support elevated levels of Myc and mRNA target supporting pluripotency and DNA repair genes hypothesized to support cellular resistance platinum DNA damage. Tumor cell intrinsic FAK kinase activity is essential for KMF tumor growth via context-dependent signaling as β-catenin activation was not sufficient to promote tumor growth in the absence of FAK.

*Figure 10 continued on next page*

*Figure 10 continued*

DOI: https://doi.org/10.7554/eLife.47327.031

The following source data is available for figure 10:

**Source data 1.** RNA sequencing annotated list of differentially expressed genes in KMF, KT13 FAK KO, FAK-WT, FAK-R454, and β-catenin ΔGSK cells grown as tumorspheres.
DOI: https://doi.org/10.7554/eLife.47327.032

**Source data 2.** List of 135 FAK-activity and β-catenin enhanced mRNAs in KMF matched to genes elevated in HGSOC (TCGA).
DOI: https://doi.org/10.7554/eLife.47327.033

*supplement 2*). Together, these results place FAK, β-catenin, and Myc within a common signaling pathway activated in ovarian cancer.

KEGG pathway analyses of FAK activity and β-catenin supported targets in HGSOC reveal conservation in *Pluripotency*, *Hippo Signaling*, and *Cell Cycle* (*Figure 10A*) which include targets supporting platinum resistance and stemness phenotypes. Quantitative PCR using independent experimental samples verified at least twofold changes in genes linked to DNA repair (*Recq4l*, *Spc24*, *Pkmyt1*, *Dscc1*, and *Rnf144b*) or pluripotency (*Wnt4*, *Klf5*, *Amigo2*, *Fos*, and *Ndufa4l2*) that were elevated by FAK-WT re-expression (*Figure 10B*). Although our studies did not establish causality of mRNA changes with FAK phenotypes supporting CP resistance and pluripotency of KMF cells, these targets are part of the FAK activity- and β-catenin-regulated 135 genes elevated in HGSOC. KMF cells are a unique murine model with profound similarities to HGSOC and will be made available to the research community.

## Discussion

Platinum-resistant ovarian carcinomas have complex tumor genomes, few targetable mutations, and no effective treatments (*Patch et al., 2015*). Gene breakage, gains, or losses are common drivers of tumor cell phenotypes. Using a new in vivo-evolved murine ovarian cancer model termed KMF - denoting gains in genes for <u>K</u>ras, <u>M</u>yc, and <u>F</u>AK – we demonstrate the functional significance of *PTK2* (FAK) gains observed in HGSOC tumors. KMF cells exhibit more aggressive tumor growth, greater tumorsphere formation in vitro, elevated FAK Y397 phosphorylation, increased β-catenin and ALDH activities, and increased resistance to cisplatin-mediated cytotoxicity compared to parental ID8 cells. In both KMF and human OVCAR3 ovarian carcinoma cells, we identify tumorsphere-associated non-canonical FAK signaling as supporting CSC phenotypes and intrinsic cisplatin resistance. This context-dependent signaling is consistent with an oncogenic role for FAK activation in ovarian cancer.

Although *MYC* amplification at 8q24.21 in HGSOC is associated with a poor prognosis (*Goode et al., 2010*), less is known about *PTK2* amplification at 8q24.3. We show that over 70% of HGSOC patient tumors contain gains at both *PTK2* and *MYC* loci, that *PTK2* copy number parallels *PTK2* mRNA and FAK protein increases, and that elevated *PTK2* mRNA levels are associated with decreased patient disease-free survival. We identify a set of 36 genes associated with *PTK2* gain predictive of HGSOC relapse. Additionally, we identify *Myc* as part of a set of 135 genes increased in murine KMF cells in a FAK *kinase-dependent* manner that also are highly expressed in HGSOC tumors. As FAK has also been proposed to function downstream of Wnt-Myc signaling in intestinal regeneration and tumorigenesis (*Ashton et al., 2010*), the contribution of FAK in support of Wnt signaling may be mediated by multiple mechanisms (*Chen et al., 2012*; *Chen et al., 2018*; *Gao et al., 2015*; *Kolev et al., 2017*). Moreover, recent studies show that FAK and β-catenin overexpression cooperate to induce hepatocellular carcinoma (HCC) in mice (*Shang et al., 2019*) and that FAK promotes CSC-like phenotypes in HCC cells (*Fan et al., 2019*).

While platinum and taxane chemotherapy kills most ovarian tumor cells, we unexpectedly find that FAK activation is elevated in the residual tumor cells of patients undergoing chemotherapy, in mouse tumors, and in isolated ovarian carcinoma tumorspheres after cisplatin chemotherapy (*Figure 10C*). Previous studies showed increased FAK Y397 phosphorylation during the processes of acquired CP resistance of cultured ovarian carcinoma cells (*Villedieu et al., 2006*). This is consistent with studies linking chemotherapy to selective CSC survival (*Wiechert et al., 2016*) and we show

that FAK inhibition compromises ALDH levels and CSC generation. Notably, platinum-resistant cells can acquire FAK dependence for growth. This dependence was manifest when culturing ovarian carcinoma cells as tumorspheres and this 3D selective phenotype has been observed in breast (*Tanjoni et al., 2010*) and squamous cell carcinoma models (*Serrels et al., 2012*). In the A2780 and A2780-CP70 models, we found selective FAK dependence for growth in the CP resistant but not parental CP-sensitive cells.

Via complementary approaches including pharmacological inhibition, FAK knockout, and FAK re-expression, we show that FAK signaling sustains both intrinsic and acquired resistance to cisplatin chemotherapy in part via promoting β-catenin activation (*Figure 10C*). Notably, a FAK to β-catenin signaling linkage functions as an adaptive chemotherapy resistance pathway in *BRAF* mutated colon cancer (*Chen et al., 2018*). Stabilized β-catenin ΔGSK expression in KMF FAK KO cells supported canonical Wnt target genes, yet β-catenin ΔGSK was unexpectedly insufficient to rescue FAK KO growth as tumors. This may be due to weak oncogenic activity of the β-catenin ΔGSK construct (*Barth et al., 1999*) or due to a supporting requirement for FAK. Additionally, the FAK-associated protein Rgnef is also essential for KMF tumorsphere growth and protection from oxidative stress (*Kleinschmidt et al., 2019*). We show that FAK expression and intrinsic activity are essential for KMF tumor growth and that elevated FAK activity and Y397 phosphorylation is an acquired and targetable cellular adaptation of cisplatin resistance in HGSOC.

In cell culture, cisplatin resistant cells acquired dependence on FAK activity to maintain proliferation as 3D tumorspheres without alterations in 2D growth. Single agent pharmacological FAK inhibition did not promote apoptosis of platinum-resistant ovarian cells. Rather, the combination of FAK inhibition (genomic and pharmacological) with cisplatin-triggered apoptosis of platinum-resistant cells as tumorspheres in vitro and prevented the growth of platinum-resistant tumors in mice. To this end, a phase I/II clinical trial for recurrent platinum-resistant ovarian cancer termed ROCKIF (Re-sensitization of platinum-resistant Ovarian Cancer by Kinase Inhibition of FAK, NCT03287271) has been initiated. ROCKIF will investigate whether the small molecule FAK inhibitor defactinib, in combination with carboplatin and paclitaxel chemotherapy, can provide benefit for this difficult to treat patient population.

# Materials and methods

## Key resources table

| Reagent type (species) or resource | Designation | Source or reference | Identifiers | Additional information |
|---|---|---|---|---|
| Antibody | anti-FAK (mouse monoclonal) | Millipore Sigma | clone 4.47; Cat# 05–537; RRID:AB_2173817 | WB (1:1000) |
| Antibody | anti-FAK (rabbit polyclonal) | Millipore Sigma | Cat# 06–543; RRID:AB_310162 | WB (1:1000) |
| Antibody | anti-phospho-FAK (Tyr397) (rabbit monoclonal) | Thermo Fischer Scientific | clone 141–9; Cat# 44–625G; RRID:AB_2533702 | WB (1:1000) |
| Antibody | anti-phospho-FAK (Tyr397) (rabbit monoclonal) | Thermo Fischer Scientific | clone 31H5L17; Cat# 700255; RRID:AB_2532307 | WB (1:1000) |
| Antibody | anti-phospho-FAK (Tyr397) (rabbit monoclonal) | Abcam | clone EP2160Y; Cat# ab81298; RRID:AB_1640500 | WB (1:1000) |
| Antibody | anti-E-cadherin (mouse monoclonal) | Cell Signaling Technology | clone 4A2; Cat# 14472; RRID:AB_2728770 | WB (1:1000) |
| Antibody | anti-β-actin (mouse monoclonal) | Millipore Sigma | clone AC-74; RRID:AB_476697 | WB (1:1000) |
| Antibody | anti-β-actin (mouse monoclonal) | Proteintech Group | Cat# 60008–1; RRID:AB_2289225 | WB (1:1000) |

*Continued on next page*

*Continued*

| Reagent type (species) or resource | Designation | Source or reference | Identifiers | Additional information |
|---|---|---|---|---|
| Antibody | anti-β-cateninXP (rabbit monoclonal) | Cell Signaling Technology | clone D10A8; Cat# 8480; RRID:AB_11127855 | WB (1:1000) |
| Antibody | anti-non-phospho (Active) β-Catenin (Ser33/37/Thr41) (rabbit monoclonal) | Cell Signaling Technology | clone D13A1; Cat# 8814; RRID:AB_11127203 | WB (1:1000) |
| Antibody | anti-β-Catenin (phospho Y142) (rabbit polyclonal) | Abcam | ab27798; RRID:AB_725969 | WB (1:1000) |
| Antibody | anti-c-Myc XP (rabbit monoclonal) | Cell Signaling Technology | clone D84C12; Cat# 5605; RRID:AB_1903938 | WB (1:1000) |
| Antibody | anti-Pyk2 (mouse monoclonal) | Cell Signaling Technology | clone 5E2; Cat# 3480; RRID:AB_2174093 | WB (1:1000) |
| Antibody | anti-p21 (mouse monoclonal) | Santa Cruz Biotechnology | clone F5; Cat# sc-6246; RRID:AB_628073 | WB (1:250) |
| Antibody | anti-GFP (mouse monoclonal) | Santa Cruz Biotechnology | clone B2; Cat# sc-9996; RRID:AB_627695 | WB (1:1000) |
| Antibody | anti-p53 (Pab 240) (mouse monoclonal) | Santa Cruz Biotechnology | Cat# sc-99, RRID:AB_628086 | WB (1:250) |
| Antibody | Anti-α-Tubulin (mouse monoclonal) | Millipore Sigma | Cat# T6199; RRID:AB_477583 | WB (1:1000) |
| Antibody | anti-ALDH1A1 (rabbit polyclonal) | Abcam | Cat# ab23375; RRID:AB_2224009 | WB (1:1000) |
| Antibody | anti-ALDH1A2 (rabbit polyclonal) | Proteintech Group | Cat# 13951–1-AP, RRID:AB_2224033 | WB (1:1000) |
| Antibody | anti-ALDH1B1 (rabbit polyclonal) | Proteintech Group | Cat# 15560–1-AP, RRID:AB_2224162 | WB (1:1000) |
| Antibody | anti-ALDH3B1 (rabbit polyclonal) | Proteintech Group | Cat# 19446–1-AP | WB (1:1000) |
| Antibody | anti-Ki67 (rabbit polyclonal) | Abcam | Cat# ab15580; RRID:AB_443209 | WB (1:1000) |
| Antibody | anti-Pax8 (rabbit polyclonal) | Proteintech Group | Cat# 10336–1-AP; RRID:AB_2236705 | WB (1:1000) |
| Antibody | anti-p53 (mouse monoclonal) | Abcam | clone PAb 240; RRID:AB_303198 | WB (1:250) |
| Antibody | anti-Cyclin D1 (rabbit polyclonal) | Cell Signaling Technology | Cat# 2922; RRID:AB_2228523 | WB (1:1000) |
| Antibody | Alexa Fluor 700 Rat Anti-Mouse CD45 | Thermo Fisher Scientific | clone 30-F11; Cat# 45-0451-80; RRID: AB_891454 | 1 ul per test |
| Strain, strain background (*Escherichia coli*) | Stellar Competent Cells, *E. coli* HST08 strain | Takara | Cat# 636763 | Chemically competent cells |
| Strain, strain background (*Escherichia coli*) | One Shot Stbl3 Chemically Competent *E. coli* | Life Technologies | Cat# C737303 | Chemically competent cells |
| Chemical compound, drug | Jet PRIME | Polyplus-transfection | Cat#114–15 | |
| Chemical compound, drug | FuGENE HD Transfection Reagent | Promega | Cat# E2311 | |

*Continued on next page*

*Continued*

| Reagent type (species) or resource | Designation | Source or reference | Identifiers | Additional information |
|---|---|---|---|---|
| Chemical compound, drug | Halt Protease Inhibitor Cocktail (100X) | Thermo Fischer Scientific | Cat# 87786 | |
| Chemical compound, drug | staurosporine | Cell Signaling Technology | Cat# 9953S | |
| Chemical compound, drug | cisplatin | Enzo Life Sciences | Cat# 89150–634 | |
| Chemical compound, drug | GSK inhibitor CHIR99021 | Millipore Sigma | Cat# SML1046 | |
| Chemical compound, drug | Propidium iodide | BioLegend | Cat# 421301 | |
| Chemical compound, drug | DTT | Bio Basic | Cat# DB0058-5 | |
| Peptide, recombinant protein | Mouse FAK (phospho Y397) peptide | Abcam | Cat# ab40145 | |
| Commercial assay, kit | PureLink RNA Mini Kit | Thermo Fisher Scientific | Cat# 12183020 | |
| Commercial assay, kit | Ambion Homogenizer | Thermo Fisher Scientific | Cat# 12183026 | |
| Commercial assay, kit | XTT Cell Proliferation Assay Kit | ATCC | Cat# 30–1011K | |
| Commercial assay, kit | Dual-Luciferase Reporter Assay System | Promega | Cat# E1910 | |
| Commercial assay, kit | PE Annexin V Apoptosis Detection Kit I | BD Biosciences | Cat# 559763 | |
| Commercial assay, kit | AlamarBlue Cell Viability Reagent | Thermo Fischer Scientific | Cat# DAL1025 | |
| Commercial assay, kit | iTaq Universal SYBR Green Supermix | Biorad | Cat# 1725121 | |
| Commercial assay, kit | High-Capacity cDNA Reverse Transcription Kit | Thermo Fisher Scientific | Cat# 4368814 | |
| Commercial assay, kit | AldeRed ALDH Detection Assay | Millipore Sigma | Cat# SCR150 | |
| Cell line (*Homo-sapiens*) | OVCAR3 | NCI Tumor Repository | RRID:CVCL_0465 | |
| Cell line (*Homo-sapiens*) | OVCAR10 | Denise Connolly (Fox Chase Cancer Center) | RRID:CVCL_4377 | |
| Cell line (*Homo-sapiens*) | OVCAR10-CP | this paper | Schlaepfer Lab | |
| Cell line (*Homo-sapiens*) | A2780 | Denise Connolly (Fox Chase Cancer Center) | RRID:CVCL_0134 | |
| Cell line (*Homo-sapiens*) | A2780-CP70 | Denise Connolly (Fox Chase Cancer Center) | RRID:CVCL_0135 | |
| Cell line (*Mus-musculus*) | ID8 | Katherine Roby (University of Kansas Medical Center) | RRID:CVCL_IU14 | |

*Continued on next page*

*Continued*

| Reagent type (species) or resource | Designation | Source or reference | Identifiers | Additional information |
|---|---|---|---|---|
| Cell line (*Mus-musculus*) | KMF (ID8-IP) | PMID: 23275034 | Schlaepfer Lab | KMF cells were isolated from peritoneal ascites of ID8-injected C57Bl6 tumor-bearing mice as described (PMID: 23275034) |
| Recombinant DNA reagent | pUltra-Chili-Luc | Addgene | Plasmid # 48688 | |
| Recombinant DNA reagent | MSCV-beta-catenin-IRES-GFP | Addgene | Plasmid # 14717 | |
| Recombinant DNA reagent | pSpCas9n(BB)—2A-Puro (PX462) | Addgene | Plasmid # 48141 | |
| Recombinant DNA reagent | M50 Super 8x TOPFlash | Addgene | Plasmid #12456 | |
| Recombinant DNA reagent | pCDH-CMV-MCS-EF1α-Puro Cloning and Expression Lentivector | System Biosciences | Cat# CD510B-1 | |
| Recombinant DNA reagent | 7TGP | Addgene | Plasmid # 24305 | |
| Recombinant DNA reagent | psPAX2 | Addgene | Plasmid #12260 | |
| Recombinant DNA reagent | pMD2.G | Addgene | Plasmid #12259 | |

## Plasmids and reagents

The dTomato with luciferase lentiviral vector, pUltra-Chili-Luc, was a gift from Malcolm Moore (Addgene #48688). The lentiviral vector MSCV-β-catenin (ΔGSK-KT3)-IRES-GFP was a gift from Tannishtha Reya (Addgene #14717). The CRISPR/Cas9 plasmid pSpCas9n(BB)—2A-Puro was a gift from Feng Zhang (Addgene #48141). M50 Super 8x TOPFlash was a gift from Randall Moon (Addgene #12456). Lentiviral vectors for murine GFP-FAK and GFP-FAK R454 in pCDH-CMV-MCS-Puro (System Biosciences) were used as described (*Chen et al., 2012*). FAKi (VS-4718) was from Verastem Inc FAKi, cisplatin (Enzo Life Sciences) or staurosporine (Calbiochem) were dissolved in DMSO for in vitro studies. VS-4718 was suspended in 0.5% carboxymethyl cellulose (Sigma) and 0.1% Tween 80 (Sigma) in sterile water and administered twice daily by oral gavage for tumor experiments. For mouse experiments, cisplatin and paclitaxel (APP Pharmaceuticals) were obtained from the Moores Cancer Center Pharmacy.

## Cells

Human ovarian carcinoma A2780, A2780-CP70, and OVCAR10 cell lines were from Denise Connolly (Fox Chase Cancer Center, PA). NIH OVCAR3 cells were from the Division of Cancer Treatment and Diagnosis Tumor Repository, National Cancer Institute (Frederick, MD), murine ovarian ID8 cells were from Katherine Roby (University of Kansas Medical Center), and KMF cells were isolated from peritoneal ascites of ID8-injected C57Bl6 tumor-bearing mice as described (*Ward et al., 2013*). Intermittent CP exposure (10 μM for 24 hr), cell recovery (7 days), and repeated exposure-recovery (five times) was used to generate OVCAR10-CP cells and maintain A2780-CP70 cells. All cells were from validated sources and were evaluated for mycoplasma contamination.

OVCAR3 FAK knockout cells were created using CRISPR/Cas9 targeting. pSpCas9n(BB)—2A-Puro was used to deliver guide RNAs (ACTGGTATGGAACGTTCTCC and TGAGTCTTAGTACTCGAATT) targeting exon 3 of human *PTK2*. Transfected cells were enriched by puromycin (1 μg/ml, 3 days) and clones selected by dilution. Loss of FAK expression was verified by immunoblotting. DNA sequencing was used to verify insertions/deletions introducing stop codons in *PTK2* exon 3. FAK re-expressing cells were generated by lentiviral transduction of OVCAR3 FAK KO clone AB21, puromycin selection, enrichment by flow cytometry, and GFP-FAK protein expression verified by immunoblotting. KMF FAK KO cells were generated by CRISPR/Cas9 targeting. pSpCas9n(BB)—2A-Puro was

used to deliver two independent guide RNAs (ACTTACATGGTAGCTCGCGG and CACTCCCA-CAGCCATCCTAT) targeting exon 4 of murine *Ptk2*. Transfected cells were enriched by puromycin selection (3.5 µg/mL for 24 hr) and clones selected by dilution. Loss of FAK expression was verified by immunoblotting. DNA sequencing was used to verify insertions/deletions introducing stop codons in murine *Ptk2* exon 4. GFP-FAK-WT, GFP-FAK-R454, and ΔGSK β-catenin (GFP expressed independently) were generated by lentiviral (for FAK) or retroviral (for β-catenin ΔGSK) transduction, puromycin or hygromycin selection, enrichment by flow cytometry, and protein expression verified by immunoblotting. For adherent 2D growth, cells were maintained in DMEM (OVCAR10, OVCAR10-CP, ID8, and KMF) or RPMI 1640 (A2780, A2780-CP70, and OVCAR3) supplemented with 10% fetal bovine serum (FBS, Omega Scientific), 1 mM non-essential amino acids, 100 U/ml penicillin, and 100 µg/ml streptomycin on cell culture-treated plastic plates (Costar). For growth as tumorspheres, cells were seeded in poly-hydroxyethyl methacrylic acid (poly-HEMA) coated Costar plates (non-adherent) in serum-free CSC media (3D Tumorsphere Medium XF, PromoCell GmbH) at cell dilutions recommended by the manufacturer. Prior to tumor initiation experiments, KMF, A2780, and A2780-CP70 cells were transduced with a lentiviral vector expressing dTomato and luciferase (pUltra-Chili-Luc) or mCherry (pCDH-CMV-MSCI) and enriched by fluorescence sorting.

## 2D and 3D cell growth assays

For 2D growth, cells were seeded ($3 \times 10^5$ cells per well) in tissue culture-treated 6-well plates (Costar). At the indicated time, cells were enumerated and stained with Trypan blue (ViCell XR, Beckman). Alternatively, cell metabolic activity was measured by a colorimetric XTT assay (Sigma). For 3D tumorspheres, cells were seeded at 10,000 cells/ml equivalent in poly-HEMA-coated 6-, 24-, or 96-well plates (Costar) for 5 days. At the indicated times, 3D tumorspheres were phase-contrast imaged (Olympus CKX41), enumerated (ViCell XR), or collected by centrifugation. Spheroid size was determined using Image J (NIH). Alternatively, cell metabolic activity was measured by a colorimetric XTT assay (Sigma). For methylcellulose colony formation, cells were suspended in 1% methylcellulose diluted in 2D growth media, $10^4$ plated in six-well poly-HEMA-coated plates, and colony formation analyzed after 21 days. Cells from methylcellulose colonies were collected by dilution-dispersion in PBS, centrifugation at 400 xg, and washed in PBS prior to enumeration or cell lysis. Cells were used at passage 10 to 35 and mycoplasma testing was performed every 3 months. For all experiments, triplicate experimental points were evaluated (technical replicates) and experiments were repeated at least two times (biological replicates).

## Patient tumor samples

De-identified human ovarian cancer tissue specimens from consented patients were obtained from the Fox Chase Cancer Center (FCCC) Biosample Repository Facility (BRF) under Institutional Review Board (IRB) approved protocols (IRB 11–866 and IRB 08–851). FCCC staff queried the BRF sample database to identify participants that received carboplatin and paclitaxel neoadjuvant chemotherapy. Biopsy specimens were obtained from FCCC Surgical Pathology, sectioned, H and E stained, and reviewed by a board-certified pathologist. FFPE blocks from the biopsy and the corresponding surgical resection blocks banked by the BRF were cut to obtain one H and E stained slide and six additional unstained sections. One section each from pre-treatment biopsy and post-neoadjuvant treatment surgical resection specimen was stained for Pax8 by the FCCC Histopathology Facility. The remainder of unstained slides were sent to UCSD for additional staining performed under UCSD IRB-approved protocol (IRB 110805).

## Cell cycle analysis

Cells were collected as a single cell suspension by limited trypsin treatment, fixed in 70% ethanol and stored at −20°C overnight. Cells were incubated in 100 µl of PBS containing DNAse-free RNAse (100 µg/mL, Qiagen). After 45 min, propidium iodide (10 µg/mL) was added prior to flow cytometry and analyzed using FlowJo (v9.5.1) and ModFit LT (Verity Software House) software.

## β-catenin transcriptional activity

Integrated reporter: 293 T cell transfection with a β-catenin DNA binding reporter (7X-TCF repeat sequence AGATCAAAGGgg) driving eGFP (7TGP, Addgene #24305, gift from Roel Nusse) was

packaged with psPAX2 (gift from Didier Trono, Addgene #12260) and pMD2.G (gift from Didier Trono, Addgene #12259) using Fugene HD (Thermo) according to the manufacturer's instructions. Media was replaced after 6 hr, conditioned media with virus collected (36 hr), centrifuged (500 g), and target cells infected for 24 hr with polybrene (8 µg/ml Sigma). Transduced cells were selected by puromycin (1 µg/ml, 5 days). Proliferating cells were seeded in 3D tumorsphere growth conditions (PromoCell), DMSO or GSK3β inhibitor (CHIR99021, Sigma, 3 µM) added after 24 hr, and cells evaluated by flow cytometry (BD FACSCelesta) after 5 days. For transient transfection of a β-catenin TOP-Flash reporter, 3.5e4 cells were seeded in triplicate in 24-well plates and co-transfected with M50 Super 8x TOPFlash with firefly and *Renilla* luciferase expression vectors using Fugene HD (Promega). After 5 hr, media was replaced and cells treated with DMSO or 3 µM GSKi (CHIR99021). After 24 hr, cells were evaluated using the Dual-Luciferase Reporter Assay system (Promega, E1910) and a plate luminometer (Dynex Tech., VA).

## Viability and cytotoxicity studies

For annexin V and 7-AAD staining, cells were cultured in adherent or non-adherent conditions as described above for 24 hr or 5 days, respectively, with increasing cisplatin treatment (0 to 100 µM). Cells were dissociated using limited trypsin treatment, washed by centrifugation, suspended in annexin-V binding buffer (10 mM Hepes, 140 mM NaCl and 2.5 mM $CaCl_2$) and incubated with allophycocyanin-conjugated Annexin V (eBioscience) and 7-aminoactinomycin D (7-AAD) for 10 min at RT prior to analysis on a FACSCalibur flow cytometer (BD Biosciences). Post-acquisition analyses were performed using CellQuest Pro (BD Biosciences) or FlowJo software. For AlamarBlue (Life Technologies) assays, cells were cultured in 96-well poly-HEMA-coated plates as above for 5 days. AlamarBlue reagent (Life Technologies) was added to each sample and incubated at 37°C at 5% $CO_2$ for 24 hr. Viability was analyzed by resorufin production via absorbance at 570/600 nm using a Synergy HTX spectrophotometer (BioTek Instruments).

## Cisplatin cytotoxicity

Cells (10,000 in 90 µl) were plated in tissue culture-treated 96-well plates (Costar). At 24 hr, increasing concentrations of cisplatin were added in growth media (10 µl), and the number of viable cells determined at 72 hr using the CellTiter 96 AQueous One Solution Cell Proliferation Assay (Promega). Measurement of cell resistance to CP-induced cytotoxicity were performed by colorimetric XTT cell staining (Sigma). In 2D culture, OVCAR10-CP and OVCAR10 cells exhibit 10.9 ± 2.2 µM and 1.4 ± 0.6 µM EC50 values to CP treatment, respectively. A2780-CP70 and A2780 cells exhibit 62.2 ± 8.7 µM and 5.6 ± 3.2 µM EC50 values to CP treatment, respectively. EC50 values were calculated using Prism (v7, GraphPad).

## ALDEFLUOR assay

The ALDEFLUOR fluorescent assay (Stemcell Technologies) was used to measure cell-associated ALDH activity. Cells were cultured as tumorspheres, treated with the indication concentrations of cisplatin or VS-4718 for 5 days, collected by centrifugation, dissociated by trypsinization, resuspended in Aldefluor assay buffer containing ALDH substrate (BODIPY-aminoacetaldehyde), and incubated for 45 min at 37°C with or without the ALDH inhibitor diethylamino-benzaldehyde (DEAB). AldeRed substrate (EMD Millipore) was used with cells expressing GFP. Individual gates were used to determine the percentage of ALDEFLUOR-positive cells per experimental point relative to DEAB-inhibitor treated controls. For analysis of ALDH activity in ascites-associated cells, pooled isolates from peritoneal washings of each experimental group were dissociated by trypsinization, treated with red blood cell lysis buffer (Biolegend), and processed as described above.

## Quantitative RT-PCR

Total RNAs were extracted using PureLink RNA Mini Kit (Thermo) and cDNA prepared using the High-Capacity cDNA Reverse Transcription Kit (Thermo) from 1 µg total RNA. Target transcripts were amplified using a LightCycler 480 (Roche Applied Science), Premix Ex Taq probe qPCR Kit, iTaq Universal SYBR Green Supermix (Bio-Rad) with cDNA template and primers *Table 2*. according to manufacturer instructions. Target gene expression was normalized to 60S ribosomal protein L19

**Table 2.** Primers used for qRT-PCR.

| Gene | Sequence |
| --- | --- |
| musRecql4F | CACCTGAGTCGAGCTGCA A |
| musRecql4R | AGCCTCTTCCCATAGTCTTGT |
| musSpc24F | AGGCTACGTCAGCTCATCAC |
| musSpc24R | ATCATCCCTGGCTCGCATTC |
| musPkmyt1F | TACCTAGGGATGCCCTGGAC |
| musPkmyt1R | CAGGCTGAGGAGGTTCCTTG |
| musDscc1F | AAGTGTGGCAGCAGAGTGTT |
| musDscc1R | TCTCTCCGCACAAATCTTGGA |
| musRnf144bF | GCAAGAACTGCAAGCACACA |
| musRnf144bF | CCCACTACCTGTGTTCGGTT |
| musWnt4 F | TGCGAGGTAAAGACGTGCTG |
| musWnt4 R | CTTGAACTGTGCATTCCGAGG |
| musKlf5F | CCGGAGACGATCTGAAACACG |
| musKlf5R | GTTGATGCTGTAAGGTATGCCT |
| musAmigo2F | GGAGGTTCAAGCTGGCTGAT |
| musAmigo2R | GATGCCTCTCAGCTGTCTCC |
| musFos F | CGGCATCATCTAGGCCCAG |
| musFos R | TCTGCTGCATAGAAGGAACCG |
| musNdufa4l2F | AAAGACACCCTGGGCTCATC |
| musNdufa4l2R | TGTAGTCGGTTGAAACGGCA |
| musRPL191F | TGATCTGCTGACGGAGTTG |
| musRPL191R | GGAAAAGAAGGTCTGGTTGGA |

DOI: https://doi.org/10.7554/eLife.47327.034

(RPL19) as a housekeeping gene control. Transcript levels were calculated using the ΔΔCT (cycle threshold) method.

## Protein analyses

Protein extracts of cells were prepared using a lysis buffer containing (25 mM HEPES, pH 7.5, 150 mM NaCl, 10% glycerol, 10 mM $MgCl_2$, 1 mM EDTA, 10 mM NaF, 1 mM $Na_3VO_4$) with 1% NP-40, 0.25% sodium deoxycholate, 0.1% SDS and protease inhibitors (Roche Diagnostics). Tumors were homogenized in lysis buffer without sodium deoxycholate using Precellys24 (Bertin Instruments) bead disruption. Total protein levels in lysates were determined through a bicinchoninic acid assay (Pierce), proteins were resolved by SDS-PAGE (NuPAGE 4–12% Tris-Bis gels, Thermo), and transferred to polyvinylidene difluoride membranes (Immobilon-FL, Millipore) for immunoblotting. Levels of protein expression and/or phosphorylation were detected with specific primary antibodies and IRDye 680 goat anti-mouse and IRDye 800 goat anti-rabbit secondary antibodies. Protein bands were visualized and quantified using the Odyssey Infrared Imaging System (Li-Cor Biosciences). Alternatively, HRP-conjugated secondary antibodies were visualized by chemiluminescence detection (ChemiDoc, BioRad).

## Tumor growth in mice

All animal experiments were performed in accordance with The Association for Assessment and Accreditation for Laboratory Animal Care guidelines and approved by the UCSD Institutional Animal Care and Use Committee (S07331). A2780 or A2780-CP70 tumor growth was evaluated by IP injection of 4 million pChili-Luciferase-labeled cells mixed with Matrigel into 9-week-old female NOD SCID gamma mice (Jackson Laboratory). IVIS imaging (Day 4, 11, 18, and 23) was used to monitor tumor growth. On Day 5, mice were randomized to a control (saline injection); chemotherapy group

(CPT) receiving IP injection of cisplatin (3 mg/kg) plus paclitaxel (2 mg/kg) at Day 5, 12, and 19; VS-4718 FAK inhibitor (100 mg/kg) via oral gavage twice daily (BID); or CPT plus FAK inhibitor treatment. At Day 24, mice were euthanized, omental tumors excised, and remaining peritoneal metastatic sites quantified by dTomato fluorescence using an OV100 Small Animal Imaging Station (Olympus) and ImageJ software.

For KMF intraperitoneal tumor growth, cells were transduced with a lentiviral vector expressing dTomato and luciferase (pUltra-Chili-Luc) and were enriched by FACS. Cells were mixed with PBS + 50% Matrigel (Growth factor reduced, Corning) for a final concentration of $4 \times 10^6$ cells per 200 µL for injection in 10-week-old C57Bl/6N mice (Charles River). Tumor growth was monitored via luciferase bioluminescent imaging (IVIS, Perkin Elmer). At the indicated times, ascites-associated cells were recovered by peritoneal washings by injection and immediate removal of PBS (5 ml), followed by erythrocyte lysis (RBC lysis buffer, eBioscience), Accutase (Corning) treatment for cell dissociation, total cell enumeration (ViCell XR, Beckman) and trypan blue staining (viability >95%). Flow cytometry (BD LSRFortessa) was used to identify dTomato+ and CD45-negative (rat anti-mouse CD45, clone 30-F11, BD Biosciences) tumor cells.

## Exome sequencing and CNV analysis

Exome sequencing was performed by Novogene (Beijing, China), using genomic DNA (1 µg) isolated from ID8 or KMF cells. Genomic DNA was sheared into 180–280 bp fragments using a Covaris Sonicator (Covaris). Exome enrichment and sequencing libraries were generated using Agilent SureSelect Mouse All Exon kit (Agilent Technologies) following manufacturer's recommendations. Each exome was sequenced using a 150 bp paired-end protocol on the Illumina HiSeq platform, generating 47M reads for the ID8 sample and 61M reads for the KMF sample. (https://software.broadinstitute.org/gatk/best-practices). Reads were aligned with BWA MEM 0.7.12 (*Li and Durbin, 2009*) to mouse genome GRCm38_68. Variants were called with GATK 3.4 according to the Broad Institute's best practices (https://software.broadinstitute.org/gatk/best-practices) (*McKenna et al., 2010*). Processing after alignment was carried out with SAMtools v.1.1 (*Li and Durbin, 2009*). Variants were annotated with ANNOVAR (*Wang et al., 2010*). Copy number variants were called from the same alignments with CNVkit (*Talevich et al., 2016*), visualized in the Integrative Genomics Viewer (*Robinson et al., 2011*) using standard parameters, with ID8 as normal and KMF as tumor samples. Ninety percent of exons were sequenced at 100X.

## RNA sequencing and analyses

Total RNA was isolated from cells growing in suspension using PureLink RNA Mini Kit (Thermo Fisher). Three independent samples of RNA were isolated from ID8 or KMF cells grown in 3D Promo-Cell XF media as tumorspheres for 5 days at various cell passages. RNA sequencing was performed by Novogene (Beijing, China). Three replicate RNA samples were obtained from KT13 FAK KO, GFP-FAK WT, GFP-FAK R454 (kinase-inactive), and FAK KO expressing a ∆GSK β-catenin. RNA library preparation was performed using NEB Next Ultra RNA Library Prep Kit (New England Biolabs). Each transcriptome was sequenced using a 150 bp paired-end protocol on the Illumina HiSeq platform. At least 60 million clean reads were generated per sample. Reads were mapped (>90%) to the reference genome using TopHat2 (*Kim et al., 2013*). Novogene analyses used ClusterProfiler software for enrichment analysis, including GO Enrichment, DO Enrichment, KEGG and Reactome database enrichment to analyze and visualize functional profiles of genomic coordinates, genes and gene clusters. Novogene performed differential expression analysis of two conditions/groups by using the DESeq2 R package. Clustering and grouping analyses used transcripts with FPKM values > 1 and an adjusted p value < 0.05. Each dataset was subject to Gene Set Enrichment Analysis (GSEA) and The Molecular Signatures Database (MSigDB) analysis. The murine FAK activity and β-catenin targets were compared with the total list of all genes gained in HGSOC by Genomic Identification of Significant Targets in Cancer (GISTIC, ov_tcga).

## Ovarian cancer *PTK2* transcriptomic survival analysis

Survival analysis was performed using a database of ovarian cancer samples (*Pénzváltó et al., 2014*). The TCGA dataset was used to link copy number gains to gene expression (*Cancer Genome Atlas Research Network, 2011*). Samples with copy number gains were designated into one cohort

and all remaining samples were designated into a second cohort. Gene expression was compared between cohorts using a non-parametric Mann-Whitney test. Genes with a fold change over two and a p-value below 1E-04 were accepted as statistically significant. The mean expression of all significant genes was computed for each sample and was used in subsequent analyses for the selected gene. Cox proportional hazards regression was performed for relapse-free survival and for overall survival. Correlation between mRNA expression and survival was assessed using the Kaplan-Meier plotter (*Gyorffy et al., 2012*) for *PTK2* mRNA levels in 1435 annotated ovarian cancer patient samples. Selections were: relapse-free survival, split patients by median, stage (all), histology (serous), grade (all), debulk (all), and chemotherapy treatments (all).

## Proteomics

ECM-enriched protein extracts from tumorsphere cultures in PromoCell were prepared by trypsin digestion as described (*Ojalill et al., 2018*). Peptides were separated by a nanoflow HPLC system (Easy-nLC1000, Thermo) coupled to a Q Exactive Hybrid Quadrupole-Orbitrap Mass Spectrometer (Thermo). A full MS (mass spectrum) scan over the mass-to-charge (m/z) range of 300–2000 with a resolution of 140,000 followed by data-dependent acquisition of with an isolation window of 2.0 m/z and a dynamic exclusion time of 20 s was performed. The top 10 ions were fragmented by higher energy collisional dissociation (HCD) with a normalized collision energy of 27 and scanned over the m/z range of 200–2000 with a resolution of 17,500. After the MS2 scan for each of the top 10 ions had been obtained, a new full MS scan was acquired and the process repeated until the end of the 70 min run. Three repeated runs per sample were performed. Tandem mass spectra were searched using MaxQuant software (v1.5.2.8) against reviewed (SwissProt) mouse sequences of UniProtKB release 2018_08. Peptide-spectrum-match- and protein-level false discovery rates were set at 0.01. Carbamidomethyl (C), as a fixed modification, and oxidation (M, P, K) as dynamic modifications were included.

A maximum of two missed cleavages was allowed. The LC-MS profiles were aligned, and the identifications were transferred to non-sequenced or non-identified MS features in other LC-MS runs (matching between runs). The extracted ion intensities of all peptides matching to the same protein from the three technical replicates were summed. Proteins were determined as detected in sample identification was derived from at least two unique peptide identifications. Contaminant proteins (according to the contaminants listed in MaxQuant), reverse identifications, and identifications only by site were removed. Only the proteins containing 'cell membrane', 'plasma membrane', 'cell surface', 'extracellular matrix' or 'secreted' in the cellular component gene ontology or in the subcellular location definition in the UniProt database were included in the final list. Samples were normalized by sum of protein intensities.

## Immunohistochemistry

Mouse tumors were divided into thirds and either processed for protein lysates, fixed in formalin, or frozen in optimal cutting temperature compound. For immunohistochemical staining, paraffin-embedded tumors were sectioned, mounted onto glass slides, deparaffinized, rehydrated, processed for antigen retrieval, and peroxidase quenched as described (*Tancioni et al., 2014*). Tissues were blocked (PBS with 1% BSA, and 0.1% Triton X-100) for 45 min at room temperature and incubated with anti-PAX8 (1:200), anti-FAK (1:200), anti-Ki67 (1:500), anti-active β-catenin (1:800) or anti-pY397 FAK (1:100) in blocking buffer overnight. FAK pY397 antibodies were pre-incubated with 200-fold molar excess of FAK pY397 peptide (Abcam) for 12 hr at RT prior to use in IHC staining. Processing with biotinylated goat-anti-rabbit or goat-anti-mouse IgG, Vectastain ABC Elite, and diaminobenzidine were used to visualize antibody binding. Slides were counterstained with hematoxylin. Colon or breast carcinoma tumor samples were used as controls for active β-catenin staining. High-resolution digital scans were acquired (Aperio CS2 scanner) using Image Scope software (Leica Biosystems). Images were also acquired using an upright microscope (Olympus BX43) with a color camera (Olympus SC100). A board-certified pathologist evaluated H and E, Pax8, pY397 FAK, or Ki67 stained images of patient tumor samples in a blinded manner. Quantification was performed using Aperio Image Analysis software (v12.3.0.5056) using the positive pixel count (v9) algorithm. Pax8-positive regions were identified and then these regions were manually-transposed onto images from

FAK pY397-stained serial section slides. Average intensity (I-Avg) values were obtained and percent FAK pY397 was calculated.

Frozen tumors were thin sectioned (7 μm) using a cryostat (Leica), mounted onto glass slides, fixed with acetone (or with 4% paraformaldehyde) for 10 min, permeabilized (PBS with 0.1% Triton) for 1 min, and blocked (PBS with 8% goat serum) for 2 hr at room temperature. Sections were incubated in anti-ALDH1A1 (1:100) or anti-pY397 FAK (1:100) in PBS with 2% goat serum overnight. Antibody binding was detected with goat anti-rabbit conjugated with Alexa Fluor-488. Cell nuclei were visualized using Hoechst 33342 stain (Thermo). Images were sequentially captured at 20X magnification (UPLFL objective, 1.3 NA; Olympus) using a monochrome charge-coupled camera (ORCA ER; Hamamatsu), an inverted microscope (IX81; Olympus), and Slidebook software (Intelligent Imaging). Images were pseudo-colored, overlaid, merged using Photoshop (Adobe), and quantified using Image J.

## Statistics

Statistical difference between groups was determined using one-way or two-way ANOVA with Tukey, Bonferroni's or Fisher's LSD post-hoc analysis. Differences between pairs of data were determined using an unpaired two-tailed Student's t test. For the IHC analysis the differences between pairs of data were calculated using a paired two-tailed Student's t test. All statistical analyses were performed using Prism (GraphPad Software, v7). p-Values of $<0.05$ were considered significant.

## Acknowledgements

We thank our colleagues for helpful discussion and comments. We thank R Winters (FCCC BRF manager) and D Flieder (FCCC Pathology) for identifying, reviewing the cases, and for procuring patient tumor samples used in this study. L Bean, K Anderson, K Taylor, and A Barrie are fellows of the UCSD Reproductive Medicine Gynecologic Oncology program. K Taylor and A Barrie are Gaines Gynecologic Oncology Fellows. D Connolly thanks The Roberta Dubrow Fund and The Main Line Chapter of the Board of Associates for charitable support. D Schlaepfer, M McHale, and D Stupack thank Joan Wyllie and 9 Girls Ask for charitable support.

## Additional information

### Competing interests

Vihren N Kolev: Former employee at Verastem Inc. David T Weaver, Jonathan A Pachter: employee of Verastem Inc. The other authors declare that no competing interests exist.

### Funding

| Funder | Grant reference number | Author |
| --- | --- | --- |
| National Institutes of Health | RO1 CA180769 | David D Schlaepfer |
| National Institutes of Health | RO1 CA102310 | David D Schlaepfer |
| National Institutes of Health | RO1 CA107263 | Dwayne G Stupack |
| National Institutes of Health | T32 CA121938 | Carlos J Diaz Osterman Florian J Sulzmaier |
| National Institutes of Health | P30 CA023100 | Alfredo Molinolo |
| National Institutes of Health | UL1 TR001442 | Kathleen M Fisch |
| National Institutes of Health | T36 GM095349 | Edward A Cordasco |
| National Institutes of Health | P30 CA006927 | Denise C Connolly |
| National Institutes of Health | RO1 CA195723 | Denise C Connolly |
| United States Department of Defense | W81XWH-16-1-0142 | Denise C Connolly |
| United States Department of Defense | W81XWH-19-1-0134 | David D Schlaepfer Dwayne G Stupack |

| Medical Research Council of Hungary | NVKP_16-1-2016-0037 | Balázs Győrffy |
|---|---|---|
| Medical Research Council of Hungary | 2018-1.3.1-VKE-2018-00032 | Balázs Győrffy |
| Medical Research Council of Hungary | KH-129581 | Balázs Győrffy |

The funders had no role in study design, data collection and interpretation, or the decision to submit the work for publication.

## Author contributions
Carlos J Diaz Osterman, Conceptualization, Formal analysis, Investigation, Visualization, Methodology, Writing—original draft; Duygu Ozmadenci, Conceptualization, Formal analysis, Validation, Investigation, Methodology, Writing—review and editing; Elizabeth G Kleinschmidt, Conceptualization, Formal analysis, Validation, Investigation, Methodology, Writing—original draft; Kristin N Taylor, Conceptualization, Resources, Formal analysis, Validation, Investigation, Methodology, Writing—original draft; Allison M Barrie, Resources, Formal analysis, Validation, Investigation, Methodology; Shulin Jiang, Validation, Investigation, Visualization; Lisa M Bean, Conceptualization, Formal analysis, Validation, Investigation, Visualization, Methodology; Florian J Sulzmaier, Conceptualization, Resources, Formal analysis, Investigation, Visualization, Methodology; Christine Jean, Formal analysis, Investigation, Methodology; Isabelle Tancioni, Vihren N Kolev, Validation, Investigation, Methodology; Kristen Anderson, Resources, Investigation, Visualization; Sean Uryu, Validation, Investigation; Edward A Cordasco, Investigation; Jian Li, Xiao Lei Chen, Formal analysis, Supervision, Validation, Investigation; Guo Fu, Supervision, Validation, Project administration; Marjaana Ojalill, Pekka Rappu, Formal analysis, Validation, Investigation; Jyrki Heino, Formal analysis, Validation, Project administration; Adam M Mark, Guorong Xu, Data curation, Formal analysis, Validation, Investigation; Kathleen M Fisch, Conceptualization, Resources, Supervision, Funding acquisition, Validation, Project administration, Writing—review and editing; David T Weaver, Jonathan A Pachter, Conceptualization, Supervision, Project administration; Balázs Győrffy, Conceptualization, Data curation, Formal analysis; Michael T McHale, Resources, Formal analysis, Project administration; Denise C Connolly, Conceptualization, Resources, Funding acquisition, Investigation, Methodology, Project administration; Alfredo Molinolo, Resources, Data curation, Formal analysis, Funding acquisition, Validation, Investigation, Methodology, Project administration; Dwayne G Stupack, David D Schlaepfer, Conceptualization, Resources, Data curation, Formal analysis, Supervision, Funding acquisition, Validation, Investigation, Visualization, Methodology, Writing—original draft, Project administration, Writing—review and editing

## Author ORCIDs
Pekka Rappu [ID] http://orcid.org/0000-0002-5068-2842
Dwayne G Stupack [ID] https://orcid.org/0000-0003-4396-5745
David D Schlaepfer [ID] https://orcid.org/0000-0003-4814-9210

## Ethics
Human subjects: De-identified human ovarian cancer tissue specimens from consented patients were obtained from the Fox Chase Cancer Center (FCCC) Biosample Repository Facility (BRF) under Institutional Review Board (IRB) approved protocols (IRB 11-866 and IRB 08-851). Slides were sent to UCSD for additional staining performed under UCSD IRB-approved protocol (IRB 110805).
Animal experimentation: All animal experiments were performed in accordance with The Association for Assessment and Accreditation for Laboratory Animal Care guidelines and approved by the UCSD Institutional Animal Care and Use Committee (S07331).

## Decision letter and Author response
Decision letter https://doi.org/10.7554/eLife.47327.043
Author response https://doi.org/10.7554/eLife.47327.044

# Additional files

## Supplementary files

• Transparent reporting form
DOI: https://doi.org/10.7554/eLife.47327.035

## Data availability

The exome sequencing FASTA files have been deposited to the NCBI Sequence Read Archive under accession number SRP194638. The RNA-Sequencing FASTQ files have been deposited to the NCBI Gene Expression Omnibus under the accession number GSE129099. The mass spectrometry proteomics data have been deposited to the ProteomeXchange Consortium via the PRIDE (Perez-Riverol et al., 2019) partner repository with the dataset identifier PXD013062.

The following datasets were generated:

| Author(s) | Year | Dataset title | Dataset URL | Database and Identifier |
|---|---|---|---|---|
| Marjaana Ojalill, David D Schlaepfer | 2019 | FAK activity sustains intrinsic and acquired ovarian cancer resistance to platinum chemotherapy | http://proteomecentral. proteomexchange.org/ cgi/GetDataset?ID= PXD013062 | ProteomeXchange, PXD013062 |
| David D Schlaepfer, Dwayne G Stupack | 2019 | FAK activity sustains intrinsic and acquired ovarian cancer resistance to platinum chemotherapy | https://www.ncbi.nlm. nih.gov/sra?term= SRP194638 | NCBI Sequence Read Archive, SRP194638 |
| David D Schlaepfer, Dwayne G Stupack | 2019 | FAK activity sustains intrinsic and acquired ovarian cancer resistance to platinum chemotherapy | https://www.ncbi.nlm. nih.gov/geo/query/acc. cgi?acc=GSE129099 | NCBI Gene Expression Omnibus, GSE129099 |

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
