## [Decision Letter]

Thank you for submitting your article "FAK activity sustains intrinsic and acquired ovarian cancer resistance to platinum therapy" for consideration by *eLife*. Your article has been reviewed by three peer reviewers, one of whom is a member of our Board of Reviewing Editors, and the evaluation has been overseen by Päivi Ojala as the Senior Editor. The reviewers have opted to remain anonymous.

The reviewers have discussed the reviews with one another and the Reviewing Editor has drafted this decision to help you prepare a revised submission. We are glad to say that all reviewers are happy to accept the paper in principle, subject to revisions to focus on the core important message that is novel, i.e. that high grade serous ovarian cancer (HGSOC) can acquire, and depend on, FAK that promotes cisplatin resistance – likely via a Wnt/β-catenin-Myc 'stemness' pathway. These findings were supported by the isolation and characterisation of a new mouse ovarian cancer cell line that displays gains in the Kras, Myc, and FAK genes, selected for more 'aggressive' behaviour.

The reviewers suggest some changes to focus the paper for publication in *eLife*, improving a few experiments. Major figures/text changes are requested to reduce the volume of data contained therein – so as to focus on the main core novel finding(s). Recommended is 6 to 7 figures and no more than the same number of supplementary figures (down from 17). If the number of panels per figure could also be reduced, this would benefit the manuscript for the readership of *eLife*. Suggestions for focus to provide a more succinct paper, and other reviewers' points, are outlined below.

Essential revisions:

1) The amount of data presented should be reduced and focused on the core key message and areas of novelty. Some of the data was considered more peripheral to the major novel findings and might be removed, including the p53 data that the reviewers felt would require substantial more data prior to publication. Also, the experiments with the inhibitors that led to the mid-manuscript model in Figures 5D and E – which provide slightly weaker conclusions – might be removed. Some of the genomic data may also be considered peripheral and removed if not directly relevant to the core message.

2) Whilst this manuscript does explore how FAK-dependent functions contribute to platinum resistance in the context of ovarian cancer – which is an important clinical problem – could the authors introduce some further citing of relevant literature. For example, there are already some papers that are relevant to a number of the concepts (albeit in other systems) in this manuscript that might be cited e.g. FAK and CSCs (e.g. Kolev et al., 2017), FAK and platinum resistance in ovarian cancer (e.g. Villedieu et al., 2006), FAK-dependent regulation 3D growth (versus 2D growth)/cyclin-D1/cell cycle (e.g. Serrels et al., 2012), FAK signalling relationship with Wnt/β-catenin/Myc in other contexts (e.g. Ashton et al., 2010).

Specific experimental points:

3) Regarding markers for stemness, a core 'mechanistic' theme of this manuscript is that development of platinum resistance is accompanied by acquisition of a CSC-like phenotype and resultant sensitivity to FAK kinase inhibitors. The concept of chemotherapy treatment enriching for CSCs that respond to FAK inhibition could be important when identifying patients who might respond better to FAK inhibitors. It would therefore be beneficial to be sure of the stem cell nature by some further characterisation. Since the authors base their conclusions here solely on the expression of ALDH and Tumor Initiation Frequency, both of which are used to study CSCs, the conclusions would be enhanced by testing additional markers associated with stemness.

4) Figure 1, subsection “KMF cells exhibit enhanced CSC phenotypes and cisplatin resistance”: the authors show that β-catenin levels are down in KMF cells relative to ID8 cells, but that transcriptional activity is up. They also show in Figure 3 that cisplatin resistant A2780 and OVCAR10 cells have elevated phospho-Y397 FAK and less β-catenin. Is β-catenin being redistributed to the nucleus of cisplatin resistant cells, and is this in any way regulated by FAK?

5) In Figure 4D, the authors present a secondary tumor initiation frequency study using cells isolated from either vehicle or VS-4718 treated mice. The figure legend says that it is using a dilution series of KMF cells. However, the accompanying description of this experiment in the Materials and methods does not state that the cherry-labelled KMF cells were isolated/sorted away from the other cell types that will be present within the dissociated tumor prior to injection into mice. If indeed the KMF cells were sorted, could the text be changed in the Materials and methods section to reflect this. If not, then the experiment is hard to interpret as only KMF cells sorted from tumors, as the other cell types being co-injected would have an impact on tumor growth and a particular number of live cells may not contain the same number of cancer cells. This could result in findings more difficult to interpret.

6) Figure 4E-G: in the experiment where KMF tumors were treated with CP and paclitaxel, n numbers should be increased, especially for the vehicle group.

---

## [Author Response]

Essential revisions:1) The amount of data presented should be reduced and focused on the core key message and areas of novelty. Some of the data was considered more peripheral to the major novel findings and might be removed, including the p53 data that the reviewers felt would require substantial more data prior to publication. Also, the experiments with the inhibitors that led to the mid-manuscript model in Figures 5D and E – which provide slightly weaker conclusions – might be removed. Some of the genomic data may also be considered peripheral and removed if not directly relevant to the core message.

We support these recommended changes. We have removed several figure panels, moved original Figure 5 data showing reconstitution of OVCAR3 cells to a figure supplement, removed several supplementary figures including genomics, bioinformatic analyses, and supplementary data tables (all sequencing and proteomics source data is available online). In decreasing the number of data panels per figures, we also increased the total figure number to ten. We found that this was the best flow for text and figures within a 5000 word limit for Introduction, Results, and Discussion. A number of different research groups contributed to our study, and the structure of the revised manuscript reflects everyone’s efforts.

2) Whilst this manuscript does explore how FAK-dependent functions contribute to platinum resistance in the context of ovarian cancer – which is an important clinical problem – could the authors introduce some further citing of relevant literature. For example, there are already some papers that are relevant to a number of the concepts (albeit in other systems) in this manuscript that might be cited e.g. FAK and CSCs (e.g. Kolev et al., 2017), FAK and platinum resistance in ovarian cancer (e.g. Villedieu et al., 2006), FAK-dependent regulation 3D growth (versus 2D growth)/cyclin-D1/cell cycle (e.g. Serrels et al., 2012), FAK signalling relationship with Wnt/β-catenin/Myc in other contexts (e.g. Ashton et al., 2010).

The studies listed above have been cited and are briefly discussed in the revised manuscript with relevance to the KMF model and FAK signaling.

Specific experimental points:3) Regarding markers for stemness, a core 'mechanistic' theme of this manuscript is that development of platinum resistance is accompanied by acquisition of a CSC-like phenotype and resultant sensitivity to FAK kinase inhibitors. The concept of chemotherapy treatment enriching for CSCs that respond to FAK inhibition could be important when identifying patients who might respond better to FAK inhibitors. It would therefore be beneficial to be sure of the stem cell nature by some further characterisation. Since the authors base their conclusions here solely on the expression of ALDH and Tumor Initiation Frequency, both of which are used to study CSCs, the conclusions would be enhanced by testing additional markers associated with stemness.

Canonical Wnt signaling via β-catenin is one of the ten major cancer signaling pathways (Sanchez-Vega et al., 2018) and Myc is a target of β-catenin and functions as a major pluripotency-driving transcription factor (Shang et al., 2017). We identify FAK expression and activity as essential for increased Myc and common β-catenin induced genes in the KMF model that were matched to 135 differentially-expressed genes in human serous ovarian cancer (Figure 10A). We provide additional experimental results from qRT-PCR analyses that FAK expression elevates (2 to 10-fold) several novel targets linked to pluripotency (WNT4, KLF5, AMIGO2, FOS, and NDUFA4L2) and DNA repair (RECQ4L, SPC24, PKMYT1, DSCC1, RNF144B) phenotypes in other cell systems (Figure 10B). We continue to investigate the significance of these and other mRNA changes induced by FAK. We have revised the text as follows: “Consistent with studies linking cisplatin resistance to ovarian CSC survival (Wiechert et al., 2016), FAK inhibition compromises ALDH levels and CSC phenotype generation.”

4) Figure 1, subsection “KMF cells exhibit enhanced CSC phenotypes and cisplatin resistance”: the authors show that β-catenin levels are down in KMF cells relative to ID8 cells, but that transcriptional activity is up. They also show in Figure 3 that cisplatin resistant A2780 and OVCAR10 cells have elevated phospho-Y397 FAK and less β-catenin. Is β-catenin being redistributed to the nucleus of cisplatin resistant cells, and is this in any way regulated by FAK?

We provide additional new immunoblotting data showing that the observed “lower” levels of β-catenin protein in KMF versus ID8 tumorspheres are reversed upon addition of GSK3β inhibitor (Figures 2C and D). GSK3β inhibitor did not alter β-catenin protein levels in ID8 cells. However, treatment of KMF tumorspheres with GSK3β inhibitor increased steady-state β-catenin levels. Accordingly, we find β-catenin transcriptional reporter activity elevated in KMF cells (Figure 2D and Figure 8A) and this activity is dependent on FAK expression (Figure 8A). [No comparisons of β-catenin-levels were presented in the original manuscript for A2780 OVCAR10 or paired CP-resistant cells].

5) In Figure 4D, the authors present a secondary tumor initiation frequency study using cells isolated from either vehicle or VS-4718 treated mice. The figure legend says that it is using a dilution series of KMF cells. However, the accompanying description of this experiment in the Materials and methods does not state that the cherry-labelled KMF cells were isolated/sorted away from the other cell types that will be present within the dissociated tumor prior to injection into mice. If indeed the KMF cells were sorted, could the text be changed in the Materials and methods section to reflect this. If not, then the experiment is hard to interpret as only KMF cells sorted from tumors, as the other cell types being co-injected would have an impact on tumor growth and a particular number of live cells may not contain the same number of cancer cells. This could result in findings more difficult to interpret.

In further considering these studies, we support the reviewer’s point that tumors grown in syngeneic mice contain both tumor and stromal cells. Moreover, pharmacological FAK inhibition can alter the cellular tumor microenvironment (Serrels et al, 2017). As cherry-labeled KMF cells were not sorted after dissociation and enumeration prior to secondary limited dilution tumor forming assays, we have removed the results from this in vivo experiment including the accompanying ALDH analyses. This decision was primarily based upon the general editorial request to reduce the scope and increase the focus of the revised manuscript. It is our contention that the original results of a difference in tumor initiating frequency (greater than 75-fold) would likely remain significant (original P = 4e-7 value) even if there was up to a 50% difference in tumor versus stromal number of cells within vehicle or FAKi treated tumor-bearing mice. The original data and methods were pre-posted prior to review and remain available for viewing at bioRxiv 594184 (Díaz Osterman et al., 2019).

6) Figure 4E-G: in the experiment where KMF tumors were treated with CP and paclitaxel, n numbers should be increased, especially for the vehicle group.

The goal of this experiment was to simply to demonstrate that changes in FAK Y397 phosphorylation could occur after cisplatin plus paclitaxel (CPT) chemotherapy to tumor-bearing mice, and that four pooled populations of re-isolated KMF cells from CPT-treated KMF tumor-bearing mice exhibited constitutive changes in FAK Y397 phosphorylation and CP resistance ex vivo. Results from this experiment were removed as part of the general editorial request to reduce the scope and increase the focus of the revised manuscript.